**Evolution of OH reactivity in NO-free volatile organic compound photooxidation**
**investigated by the fully explicit GECKO-A model**
Zhe Peng[1], Julia Lee-Taylor[1,2], Harald Stark[1,3], John J. Orlando[2], Bernard Aumont[4] and Jose L. Jimenez[1]
[1] Department of Chemistry and Cooperative Institute for Research in Environmental Sciences,
University of Colorado, Boulder, Colorado 80309, USA
[2] Atmospheric Chemistry Observation and Modeling Laboratory, National Center for Atmospheric
Research, Boulder, Colorado 80307, USA
[3] Aerodyne Research Inc., Billerica, Massachusetts 01821, USA
[4] Univ Paris Est Créteil and Université de Paris, CNRS, LISA, F-94010 Créteil, France
Correspondence: Zhe Peng (zhe.peng@colorado.edu) and Jose L. Jimenez
(jose.jimenez@colorado.edu)
**Abstract.** OH reactivity (OHR) is an important control on the oxidative capacity in the atmosphere but
remains poorly constrained in many environments, such as remote, rural, and urban atmospheres, as
well as laboratory experiment setups under low-NO conditions. For an improved understanding of OHR,
its evolution during oxidation of volatile organic compounds (VOCs) is a major aspect requiring better
quantification. We use the fully explicit Generator of Explicit Chemistry and Kinetics of Organics in the
Atmosphere (GECKO-A) model to study the OHR evolution in the NO-free photooxidation of several
VOCs, including decane (an alkane), m-xylene (an aromatic), and isoprene (an alkene). Oxidation
progressively produces more saturated and functionalized species. Total organic OHR (including
precursor and products, $OHR_{VOC}$) first increases for decane (as functionalization increases OH rate
coefficients), and m-xylene (as much more reactive oxygenated alkenes are formed). For isoprene, C=C
bond consumption leads to a rapid drop in $OHR_{VOC}$ before significant production of the first main
saturated multifunctional product, i.e., isoprene epoxydiol. The saturated multifunctional species in the
oxidation of different precursors have similar average $OHR_{VOC}$ per C atom. The latter oxidation follows
a similar course for different precursors, involving fragmentation of multifunctional species to eventual
oxidation of C1 and C2 fragments to $CO_2$, leading to a similar evolution of $OHR_{VOC}$ per C atom. An upper
limit of the total OH consumption during complete oxidation to $CO_2$ is roughly 3 per C atom. We also
explore the trends in radical recycling ratios. We show that differences in the evolution of $OHR_{VOC}$
between the atmosphere and an environmental chamber, and between the atmosphere and an
oxidation flow reactor (OFR) can be substantial, with the former being even larger, but these differences
are often smaller than between precursors. The Teflon wall losses of oxygenated VOCs in chambers
result in large deviations of $OHR_{VOC}$ from atmospheric conditions, especially for the oxidation of larger
precursors, where multifunctional species may suffer substantial wall losses, resulting in significant
underestimation of $OHR_{VOC}$. For OFR, the deviations of $OHR_{VOC}$ evolution from the atmospheric case are
mainly due to significant OHR contribution from $RO_2$ and lack of efficient organic photolysis. The former
can be avoided by lowering the UV lamp setting in OFR, while the latter is shown to be very difficult to
avoid. However, the former may significantly offset the slowdown in fragmentation of multifunctional
species due to lack of efficient organic photolysis.

## 1    Introduction

Photooxidation is a key process altering the concentrations of trace gases in the atmosphere (Levy II, 1971; Atkinson and Arey, 2003). It is also the main contributor to the formation of $O_3$ and secondary aerosols (Haagen-Smit, 1952; Chameides et al., 1988; Hallquist et al., 2009). Both products are major tropospheric pollutants (Nel, 2005; Cohen et al., 2017) and the latter have large climate impacts (Stocker et al., 2014).

Hydroxyl radical (OH) is the primary oxidizing agent in atmospheric photooxidation (Levy II, 1971). Its atmospheric fate is governed by the species that it reacts with, i.e., OH reactants. The first-order rate constant of OH consumption by an OH reactant is often called its OH reactivity (OHR), equal to the product of the reactant concentration and second-order rate constant with OH. Total OHR ($OHR_{tot}$), i.e., the sum of OHR across all OH reactants ($OHR_{tot} = \sum_i (k_i * c_i)$), where $k_i$ and $c_i$ are the second-order rate constant with OH and concentration of the $i$th OH reactant), is the real first-order loss rate constant of OH.

OHR has been measured for over 20 years (Kovacs and Brune, 2001) in various settings, e.g., urban areas (Lu et al., 2013; Whalley et al., 2016), forested areas (Nölscher et al., 2016; Zannoni et al., 2016), and environmental chambers (Fuchs et al., 2013; Nölscher et al., 2014; Fuchs et al., 2017; Novelli et al., 2018). Despite numerous measurements and remarkable technical developments (Yang et al., 2016; Fuchs et al., 2017), a sizable fraction of total OHR in most measurements has not been chemically speciated, leading to so-called "missing reactivity" (Williams and Brune, 2015; Yang et al., 2016). Multiple studies (Nölscher et al., 2016; Whalley et al., 2016; Sato et al., 2017) have attributed missing reactivity to the highly complex mixture of intermediates and products of volatile organic compound (VOC) oxidation, most of which are oxygenated VOCs (OVOCs). Primary VOCs themselves have been found to be the largest contributor of the speciated OHR in many studies (Yang et al., 2016). In order to well understand ambient OHR, the evolution of OHR (including that from OVOCs) during primary VOC photooxidation thus needs to be investigated.

Experimentally, this can be done in environmental chambers. However, only a few such experiments have been published (Nakashima et al., 2012; Nehr et al., 2014; Nölscher et al., 2014; Sato et al., 2017), all under high-NO conditions, where the key organic radical intermediate in VOC oxidation, i.e., organic peroxy radical ($RO_2$), mainly reacts with NO. To our knowledge, no experiment of this type at low NO, where RO2 can substantially react with hydroperoxy radical (HO2), has been published so far, potentially partially due to the difficulty in experimentally ensuring that low-NO conditions are achieved in chambers (Nguyen et al., 2014). Also, many OVOCs, which may account for missing reactivity, have sufficiently low volatility to significantly partition to chamber walls (Matsunaga and Ziemann, 2010; Krechmer et al., 2016), further complicating these experiments. The OVOC wall losses also often limit operation times of chamber experiments to a few hours, after which the wall losses are so large that meaningful interpretation of experimental results would be difficult. Therefore, the highest equivalent photochemical age that can be reached in chamber experiments is also typically hours and far shorter than would be needed to explore the OHR evolution in later stages of VOC

oxidation.

Oxidation flow reactors (OFR) are an alternative to chambers with much smaller volume, shorter

residence time (and thus smaller wall losses of trace gases), and stronger oxidative capacity (Kang et
al., 2007; Brune, 2019; Peng and Jimenez, 2020). The most common version of OFR is equipped with
low-pressure Hg lamps emitting UV at 185 and 254 nm, which photolyzes water vapor, $O_2$, and $O_3$, and
generates a large amount of OH both directly and through subsequent radical reactions. High OH
concentration in OFR often leads to equivalent photochemical age of days to weeks (Li et al., 2015;
Peng et al., 2015). In principle, OFR can also be employed to explore OHR evolution in VOC oxidation.
However, OHR from VOC (OHR$_{VOC}$, from both precursor and oxidation intermediates/products, in which
we include CO as an "organic" product of VOC oxidation) can have strong impacts on oxidative capacity
(particularly OH concentration) and hence radical chemistry in OFR at both low (Li et al., 2015; Peng et
al., 2015) and high NO (Peng and Jimenez, 2017; Peng et al., 2018). Peng and Jimenez (2020) have called
for highly chemically detailed modeling of gas-phase organic chemistry in OFR to assess the impacts of
organic OH reactants on OH in a more quantitative manner.

In this study, we explore for the first time the OHR evolution in entire NO-free VOC photooxidation

processes by modeling. Since chemical mechanism incompleteness causes other models to
unsatisfactorily simulate measured OHR$_{tot}$ (Williams and Brune, 2015), we use the fully chemically
explicit model GECKO-A (Generator of Explicit Chemistry and Kinetics of Organics in the Atmosphere)
(Aumont et al., 2005). We simulate the photooxidation of different types of VOCs in the atmosphere,
in chamber, and in OFR, to find out general trends of OHR evolution in VOC oxidation and whether VOC
oxidation chemistries in chamber and OFR are representative of that in the atmosphere in terms of
OHR evolution.
**2    Methods**

Here we first discuss the VOC precursor types and conditions selected for the model cases in this

study. Then we describe the GECKO-A model and present our additional mechanism, model, and
software development required for this study.
**2.1    Model cases**

The photooxidation of an alkane (decane), an alkene (isoprene), and an aromatic (m-xylene) is

investigated under a variety of conditions without any NO. In pristine regions such as open oceans, NO
has typical concentrations on the order of 1 ppt (Wofsy et al., 2021) and hence contribute only a few
percent to RO$_2$ loss (Peng et al., 2019). For simplicity, we choose not to maintain such a low NO level in
the simulations, but to model zero-NO cases instead. The model cases are listed in Table 1: i) two cases
under ambient conditions, one with constant sunlight at solar zenith angle of 45° and the other with
diurnally-varying solar radiation and a noontime solar zenith angle of 0°; ii) six cases under typical
chamber conditions, i.e., low (10 s-1) / high (100 s-1) precursor OHR without gas-particle-wall
partitioning, with gas-particle partitioning (no wall), and with gas-particle-wall partitioning; and iii) five
cases under OFR conditions, of which two conditions resulting in significant non-tropospheric organic
photolysis (Peng et al., 2016) and one leading to remarkable deviations of RO2 fate from that in the
troposphere are not recommended in practice, but are still included for completeness since they are
similar to conditions in some literature studies (Table 1).For the UV source in chamber cases, we adopt
the spectrum of the blacklight and fluorescence light array in the University of Colorado Environmental
Chamber Facility (CU Chamber; Krechmer et al., 2017). The CU Chamber has a volume of ~20 m3, a
surface area of ~65 m2, and an estimated wall condensation timescales of ~1000 s (Krechmer et al.,
2016). The parameterization for the reversible gas-wall partitioning is taken from Krechmer et al. (2016)
with updates of Liu et al. (2019). Wall partitioning in chambers at equilibrium is a function of the
surface-to-volume ratio (Krechmer et al., 2016). The timescale to approach equilibrium is expected to
be larger in larger chambers, but still far shorter than the long experiments needed to investigate high
photochemical ages. Therefore differences in wall partitioning timescale are not important for this
study. Figure S9 of Krechmer et al. (2016) compared the CU Chamber and a few other well-known
chambers (including very large ones such as EUPHORE (Siese et al., 2001) and SAPHIR (Rohrer et al.,
2005)), showing relatively small differences (within a factor of ~2 in terms of surface-to-volume ratio).
Therefore the conclusions about wall partitioning in this study should be approximately applicable to
most chambers. The cases under ambient conditions, and chamber conditions with low / high precursor
OHR are simulated for 10, and 6 / 30 d, respectively, to encompass an equivalent photochemical age
of >10 d in each case (given a typical average ambient OH concentration of $1.5 \times 10^6$ molecules cm$^{-3}$ in
the real atmosphere (Mao et al., 2009); see Fig. 1 for the correspondence between equivalent
photochemical age and OH exposure (OH$_{exp}$, i.e., the integral of OH concentration over time)). The
simulated OFR in the present work employs the light source parametrization obtained by Li et al. (2015)
and Peng et al. (2015). UV at both 185 and 254 nm is used to generate OH, i.e., the "OFR185" mode of
operation. The residence time in the OFR is always 3 min. Wall losses in the OFR should be smaller than
in the chamber, due to reduced wall contact (Brune 2019), and are not simulated here. As several key
parameters of the chamber and OFR cases were obtained experimentally at room temperature and
atmospheric pressure in Boulder, Colorado, USA (typically 295 K and 835 mbar), for an easier
comparison, we use these values for the temperature and atmospheric pressure of all model cases.
In addition, we simulate illustrative cases of methane oxidation, under ambient and OFR
conditions (Table 1 and Section 3.1). Note that these two simulations are performed using the GECKO-
A generated mechanism (see Section 2.2) in KinSim (Peng and Jimenez, 2019), a chemical-kinetics solver
that is not GECKO-A's default, to avoid possible numerical issues in the GECKO-A internal solver, as
methane oxidation by OH is very slow (Atkinson and Arey, 2003) and very long runs are needed.To
characterize trends of OHR evolution (see Section 3.5), the ambient cases with constant sunlight are
simulated for two more alkanes, i.e., butane and heptane (Table 1). To explore the effects of UV sources
in OFR (see Section 3.4), two simulations under a typical OFR condition with an additional broad-
spectrum UV source (5 and 10000 times the chamber UV source in this study, respectively) are
performed for isoprene (Table 1).
**2.2    The GECKO-A model**
GECKO-A (Aumont et al., 2005; with updates as described by Camredon et al., 2007; Valorso et
al., 2011; Lee-Taylor et al., 2015), is an explicit chemical model which uses known mechanisms and rates
supplemented with experimentally-based structure-activity relationships to generate comprehensive
atmospheric oxidation mechanisms for organic species. The mechanisms are implemented within a box
model with a two-step solver (Verwer, 1994; Verwer et al., 1996). In mechanism generation, isomer
lumping for mechanism reduction purposes is applied to certain products with branching ratios < 1%
(here typically N-containing products, which are not relevant for our simulations).  It has a negligible
impact on the results.
The core isoprene scheme in GECKO-A is adopted from the Master Chemical Mechanism v3.3.1
(Jenkin et al., 2015), while the meta-xylene oxidation mechanism follows MCM v3.2 (Jenkin et al, 2003,
Bloss et al, 2005), typically until ring-breaking occurs, whereupon the GECKO-A mechanism generator
implements the standard SAR protocols as described by Aumont et al. (2005), Camredon et al. (2007),
and Lee-Taylor et al. (2015). Under the zero-NO conditions employed in this study, we find that, in two
of the four m-xylene reaction channels (xylenol, 17%; and MXYLO2, 4%), some product species persist
anomalously owing to lack of alternative reaction pathways in the MCM. We therefore allow GECKO-A
to apply the standard SARs to two cyclic non-aromatic products of xylenol (MXYOLO2 and MXYOLOOH
in the 51% xylenol OH-oxidation channel, see Scheme S1). We also introduce OH-oxidation of
MXYCATECH and MXY1OOH (in the 42% and 7% xylenol OH-oxidation channels), and of MXYLOOH and
MXYLAL (in the MXYLO2 channel), assuming similarity to the MCM OH-oxidation of xylenol to
MXYOLO2, and with net OH rate constants estimated using the EPA EPISuite software package (US EPA,
2012). MXYLOOH, MXCATECH and MXYLAL each yield between two and six bicyclic non-aromatic
substituted peroxy radicals, with net OH rate constants of $1.77 \times 10^{-11}$, $1.56 \times 10^{-10}$ and $8.6 \times 10^{-13}$ $cm^3$
$molecule^{-1}$ $s^{-1}$ respectively. (The MXYLOOH OH-rate also includes MXYLAL production). MXY1OOH is
assigned a substituted single-ring hydroxy-ketone product, with OH rate constant $3.26 \times 10^{-11}$ $cm^3$
$molecule^{-1}$ $s^{-1}$. The early part of the meta-xylene reaction scheme used in this work is shown in Scheme
S1.
We tested the effect of solver integration timestep length on output precision. The output
species concentrations in all simulations but for isoprene OFR (Table 1) converge well as integration
timestep decreases (Fig. S1). In the isoprene OFR test cases, the output values oscillate over a small
range (<~5%) for integration timesteps ≤ 0.01 s (Fig. S1). Since this numerical error is smaller than typical
rate constant measurement uncertainties (from ~10% to a factor of 2–3; Burkholder et al., 2015), let
alone the uncertainties related to the SARs used in GECKO-A, it is deemed acceptable for the relevant
simulations in this study. The integration timestep for each simulation in the present work is reported
in Table 1.
We allow mechanism generation to proceed through to $CO_2$ production in most cases in this
study. The only exception is for extremely low-volatility species (saturation vapor pressure < $10^{-13}$ atm)
which are considered to be completely and irreversibly partitioned to the particle phase. Particle- and
wall-phase species are no longer considered in the OHR budget, since heterogeneous oxidation is much
slower than gas-phase oxidation (e.g., George and Abbatt, 2010). Gas-particle-wall partitioning is
activated only for the chamber cases where wall effects are considered. For the ambient cases and the
chamber cases without gas-wall partitioning, gas-particle partitioning is also disabled to avoid artificial
condensation of gases into the particle phase. In environments with very low NO (e.g., remote
atmosphere), organic aerosol concentration is typically 0.2 µg m$^{-3}$ (Hodzic et al., 2020) while most major
intermediates/products have higher saturation concentrations (C*) and hence largely stay in the gas
phase. C* is calculated using the parameterization of Nannoolal et al. (2008) (default option of GECKO-
A). Although SIMPOL (Pankow and Asher, 2008) was recommended by Krechmer et al. (2016) to
estimate C* for the chamber wall partitioning treatment using their parameterization, the C* estimates
by the Nannoolal and SIMPOL parameterizations are close (generally within a factor of 2) for the species
that can reversibly partition between the gas and wall phases (C* ~ 0.1–1000 µg m$^{-3}$) in this study. This
difference is smaller than the uncertainties of the  Krechmer et al. (2016) parameterization. Therefore,
the use of the parameterization of Nannoolal et al. (2008) for C* estimation is acceptable.

Concerns have previously been expressed about non-conservation of carbon in GECKO-A

(Mouchel-Vallon et al., 2020). This has proven in the current simulations to be almost entirely due to
lack of accounting for product $CO_2$ in some handwritten reactions. We edited the handwritten isoprene
and m-xylene schemes (see above and Section 2.2.3) for carbon balance, which reduced simulation-
end carbon losses in the m-xylene and isoprene ambient cases with constant UV from 4% and 9%,
respectively, to negligible levels (<0.4%; Fig. S2).

For the current study, we have made several updates to GECKO-A, i.e., i) inclusion of key OFR-

specific radical reactions, ii) extension of the UV range considered to cover 185 and 254 nm, and iii)
updates to the NO-free m-xylene oxidation mechanism, so that GECKO-A is able to simulate OFR
chemistry and the entire process of NO-free m-xylene photooxidation (until CO/$CO_2$). We will describe
these three updates below.
**2.2.1  Key radical reactions in oxidation flow reactor**

We have added several reactions that are unimportant in the troposphere, but that are required

to fully represent the radical chemistry within the OFR (Li et al., 2015). The most important inorganic
reactions are $H_2O$ + hv (185 nm) $\rightarrow$ H + OH, $O_2$ + hv (185 nm) $\rightarrow$ 2O($^3$P), and $O_3$ + hv (254 nm) $\rightarrow$ O($^1$D)
+ $O_2$. These three reactions, together with O($^3$P) + $O_2$ + M $\rightarrow$ $O_3$ + M and O($^1$D) + $H_2O$ $\rightarrow$ 2OH, which are
already in the GECKO-A inorganic radical chemistry scheme, are responsible for the OH generation in
OFR. The OFR radical chemistry has previously been modeled in detail using KinSim (Peng and Jimenez,
2019), which was validated against experimental observations (Li et al., 2015; Peng et al., 2015). A
comparison between KinSim and GECKO-A for a range of OFR conditions shows typical agreement
between the two models within 2% for key outputs.

Due to high OH in OFR, reaction of $RO_2$ with OH is also included in mechanism generation, with

an assumed rate constant of 1x10$^{10}$ cm$^3$ molecule$^{-1}$ s$^{-1}$ (Peng et al., 2019). The products of this type of
reaction are assumed to be RO (alkoxy radical) + $HO_2$ for alkyl $RO_2$ and R (alkyl radical) + $CO_2$ + $HO_2$ for
acyl $RO_2$. Although these reactions for certain $RO_2$ may have reaction intermediates, the reactions of
the intermediates (with OH) are believed to be very fast under OFR conditions where OH is much higher
than in the atmosphere (Peng and Jimenez, 2020) and hence only the probable final products (no
intermediates) of these reactions are included in mechanism generation. The reaction of $RO_2$ with OH
is not included in the mechanisms for the ambient and chamber simulations due to low contribution of
this pathway to the $RO_2$ fate in those cases.

### 2.2.2 Organic photolysis at 185 and 254 nm

Organic photolysis is assessed in GECKO-A via a lookup table of j-values for reference
chromophores pre-calculated at different solar zenith angles with the TUV 1-D radiative transfer model
(Madronich and Flocke, 1999). The reference cross-sections used in the model generally do not cover
the UV wavelengths at which OFR operates (with narrow peaks at 185 nm and 254 nm) since they are
not tropospherically relevant. Thus it was necessary to extend to 185 nm the relevant reference
absorption cross-sections. We have done this using literature values via the Mainz UV-Vis spectral atlas
(Keller-Rudek et al., 2020) or by extrapolating the available cross-section data, using other similar
chromophores as references. Details of all cross-section extensions are given in Table S1. Where
quantum yield information was not available, we assume values of unity since photons at 185 and 254
nm are usually sufficiently energetic to make photolysis occur (Ausloos and Lias, 1971). In case of
multiple product channels for a photolyzed molecule, the branching ratios of those channels at 185 and
254 nm are estimated through extrapolation of branching ratio data from available ranges followed by
a renormalization. Finally, we apply the OFR UV spectrum within TUV to calculate OFR-relevant j-value
lookup tables.

### 2.2.3 Mechanism of NO-free m-xylene oxidation

The meta-xylene oxidation mechanism in GECKO-A follows MCM v3.2 until all aromatic, epoxy,
or bridged-peroxy rings are broken (See Scheme S1). Since the MCM was designed for typical urban
environments with abundant $NO_x$, it omits some reaction pathways for other oxidants, assuming them
to be of negligible importance. The relevant photolysis loss pathways are slow under ambient
conditions and inactive in the OFR case. This leads to persistence and accumulation of certain
hydroperoxides and their interconverting peroxy radicals under NO-free conditions. We added two NO-
free oxidation reactions to the xylenol branch of the meta-xylene oxidation scheme, Scheme S1. In the
51% branch, we allow the unsaturated bicyclic peroxide "MXYLOOH" to react with, sequentially, OH
(estimating $k_{VOC+OH} \sim 3e^{-11}$ cm$^3$ molecule$^{-1}$ s$^{-1}$) and $HO_2$ (estimating $k_{RO2+HO2} \sim 1e^{-11}$ cm$^3$ molecule$^{-1}$ s$^{-1}$), to
produce a saturated bicyclic peroxide (denoted "TT8001" in Scheme S1). In the 42% branch, we add a
competing $O_3$ reaction with the alkoxy radical "MXCATEC1O", producing an unsaturated carbonyl
alkoxy radical "1T8000" which eliminates CH3 to form the unsaturated cyclic hydroxy dicarbonyl
"TU7000". Both products are then further oxidised via the standard GECKO SARs.

### 2.3 GECKO Loader and Plotter

To allow GECKO-A outputs, which are usually highly complex and voluminous, to be explored and
visualized in detail on standard (non-UNIX) personal computers, we have developed the GECKO Loader
and Plotter based in the data-analyzing and graphic-making package Igor Pro 8.0 (WaveMetrics, Lake
Oswego, Oregon, USA). This tool assists on the rapid and detailed analysis of model-chamber/OFR
comparison studies.

Specifically, the GECKO Loader and Plotter facilitates: i) filtering the (sometimes extremely large

and finely-resolved) model results time series to examine specific characteristics, ii) identifying the most
abundant and/or influential species in each phase (gas, particle, and wall), iii) selecting species by
specific chemical identity (molecular formula, specific formula, and/or functional group identity), iv)
plotting time series of individual species and their formation/destruction rates, v) assessing and
displaying aggregated properties (volatility distribution, mass spectrum, Henry's law constant
distribution) of the product mixture and subsets thereof, and vi) calculating bulk characteristics of the
simulation ($OH_{exp}$, $OHR_{VOC}$, light intensity, elemental ratios etc.) and relating species abundances to
them.

### 3  Results and discussions

In this section, we will show the evolution of $OHR_{VOC}$ in the photooxidation of different precursors

under various conditions. To aid the presentation of this evolution for larger precursors, whose
oxidation is more complex, the oxidation of the simplest VOC, i.e., methane, will be first discussed. After
presenting the results of individual precursors, we will compare the results between conditions and
between precursors to illustrate the general trends. Along with the OHR evolution, OH recycling ratio
($\beta_1$, defined as number of OH molecules generated from organic reactions per OH consumed by
organics) and $HO_x$ (= OH + $HO_2$) recycling ratio ($\beta_2$, defined as number of OH and $HO_2$ molecules
generated from organic reactions per OH consumed by organics) will also be discussed, as they are
important parameters that may considerably affect the budget of atmospheric oxidizing agents (Stone
et al., 2012) and the $HO_2$-to-OH ratio and $RO_2$ chemistry in OFR (Peng et al., 2015, 2019).

### 3.1  Methane

To explain one of the main features in the OHR evolution in VOC photooxidation, i.e., $OHR_{VOC}$

peaking at a certain $OH_{exp}$, the oxidation of $CH_4$ is employed as an example because of its simpler
mechanism (Scheme S2). The results of this oxidation under the ambient condition show that $OHR_{VOC}$
peaks at an $OH_{exp}$ of about $1\times10^{13}$ molecules $cm^{-3}$ s (Fig. S3). As the OHR of the precursor always
decreases during its oxidation, the appearance of such a peak of $OHR_{VOC}$ before all VOCs are finally
oxidized to $CO_2$ indicates that the OHR increase from intermediates and products is faster than the OHR
decrease of the precursor. This is obviously the case for $CH_4$ oxidation, as there is no significant $CH_4$ loss
before $OH_{exp}$ ~ $10^{13}$ molecules $cm^{-3}$ s by its very slow reaction with OH (rate constant on the order of
$10^{-15}$ $cm^3$ $molecule^{-1}$ $s^{-1}$; Atkinson and Arey, 2003) and all the non-$CO_2$ intermediates/products of the
oxidation ($CH_3OOH$, $CH_3OH$, HCHO, and CO) are orders of magnitude more reactive toward OH than is
$CH_4$ (Atkinson and Arey, 2003). This large difference in precursor and intermediate/product oxidation
timescales allows the oxidations of intermediates/products (including CO, whose reaction rate constant
with OH is ~$2\times10^{-13}$ $cm^3$ $molecule^{-1}$ $s^{-1}$; Burkholder et al., 2015) to establish a steady state, whereby the
OHR of the intermediates/products is proportional to the concentration/OHR of $CH_4$. After $OH_{exp}$ ~ $10^{13}$
molecules $cm^{-3}$ s, $CH_4$ concentration decay, and consequently that of all intermediates/products,
become significant, giving the $OHR_{VOC}$ peak around $1\times10^{13}$ molecules $cm^{-3}$ s.
We also performed a simulation under a typical OFR condition. The $OHR_{VOC}$ peak also appears
around $1 \times 10^{13}$ molecules cm$^{-3}$ s in this case for the same reasons discussed above, but its height is
almost twice that of the ambient case (Fig. S3). The OHR of CO in both cases is similar, while that of
$CH_3OH$ is higher in the ambient case but those of $CH_3OOH$ and HCHO are significantly higher in the OFR
case. This is because the relative importance of the various reactions involved in $CH_4$ oxidation (Scheme
S2) depends on the conditions in each reactor.
In the OFR case, OH and $HO_2$ concentrations are ~4 and ~3 orders of magnitude higher than
typical ambient values, respectively (Peng et al., 2015). The reactions of two intermediates, $CH_3OOH$
and HCHO, with OH and the reaction of the only major $RO_2$ involved, $CH_3OO$, with $HO_2$ are much faster
than their photolysis or the self-reaction of $CH_3OO$ (Scheme S2). Neglecting organic photolysis and
$CH_3OO$ self-reaction (and thus $CH_3OH$ as a product of the latter), the mechanism of $CH_4$ oxidation can
be simplified to an OH-driven chain ($CH_4 \rightarrow CH_3OOH \rightarrow HCHO \rightarrow CO \rightarrow CO_2$) with a fast steady-state
branch on $CH_3OOH$ ($CH_3OOH \leftrightarrow CH_3OO$). For a simple chain, as we show in Appendix A, the OHR of
precursor and that of each intermediate are equal. At the $OHR_{VOC}$ peak, the OHR of HCHO and CO are
very close to that of $CH_4$, while that of $CH_3OOH$ is larger, because the branch reaction $CH_3OOH + OH \rightarrow$
$CH_3OO + H_2O$ also contributes to OHR, but does not affect the chain (and hence the OHR of the
downstream species). With such an idealized chain mechanism, the $OHR_{VOC}$ peak height equals the
precursor OHR multiplied by the number of steps needed to produce $CO_2$.
In contrast, the $OHR_{VOC}$ peak height (and its composition) in the ambient case cannot be
explained by the simple chain. Both HCHO photolysis and $CH_3OO$ self-reaction play a major role in the
oxidation in this case, and are significantly faster than the reactions in the simple chain (HCHO + OH
and $CH_3OO + HO_2$). In terms of the relationship of these two reactions with the chain, HCHO photolysis
bypasses HCHO + OH in converting HCHO to CO, while $CH_3OO$ self-reaction bypasses $CH_3OOH$, in effect
short-circuiting the involvement of OH in the oxidation and hence lowering $OHR_{VOC}$ (Appendix A). Since
the only reaction of CO is CO + OH, its OHR at the $OHR_{VOC}$ peak is essentially unaltered.
**3.2    Decane**
**3.2.1    Ambient and OFR cases**
The evolution of $OHR_{VOC}$ during the oxidation of another alkane, decane, under ambient and OFR
conditions exhibit a smaller difference and smaller peak enhancements than those in the methane
cases (Fig. 1), although the chain lengths of the decane oxidation to $CO_2$ should be much longer than
that of methane. The ambient cases with constant and diurnal solar radiation have almost the same
$OHR_{VOC}$ evolution as a function of $OH_{exp}$ (Figs. 1 and 2).
These differences from the methane cases arise because a key assumption of the simple reaction
chain model, i.e., slow precursor decay allowing intermediates/products to build up and reach a steady
state, no longer holds in decane oxidation. The main first-generation products, i.e., secondary decyl
hydroperoxides, react with OH only less-than-3-times more rapidly (in terms of the rate constant of the
whole molecule) than does decane, as the significant activation effect of the –OOH group only applies
to the $\alpha$-H, and all other H atoms in this long chain alkyl, though less reactive, can be abstracted by OH
(Kwok and Atkinson, 1995; Aumont et al., 2005). When decyl hydroperoxides are present in significant
amounts ($OH_{exp}$ ~ $5x10^{10}$ molecules $cm^{-3}$ s), decane loss is also significant (Fig. 3). Also, oxidation of
monohydroperoxides to ketones, their most likely fate (due to the activated α-H), lowers $OHR_{VOC}$, as
the oxidation removes the most activated H (Kwok and Atkinson, 1995). The multifunctional products
of further oxidation in the mid $OH_{exp}$ range (before ~$2x10^{11}$ molecules $cm^{-3}$ s), mainly have -OOH and -
CO- (Fig. 4), which do not further increase OHR substantially with respect to monohydroperoxides, for
similar reasons as the comparison of monohydroperoxides with decane. After the $OHR_{VOC}$ peak, the
precursor is largely consumed and -CH(OOH)- groups become increasingly oxidized to -CO- in both
monohydroperoxides and multifunctional hydroperoxides (Figs. 3 and 4), which rapidly decreases
$OHR_{VOC}$. Since the decane oxidation chain does not reach a steady state, it results in only limited $OHR_{VOC}$
enhancement at peak.

In the absence of steady state for the nodes (stable species) in the decane oxidation chains

(nodes far downstream insufficiently populated), organic photolysis and $RO_2$ self- and cross-reactions
only help move OHR contributors to downstream nodes, but do not significantly change their total
concentrations. This is shown by the relatively small differences in the composition of stable OHR
contributors between the ambient and OFR cases (Fig. 3). The remarkable difference between these
cases is the contribution of $RO_2$ to OHR, which is as high as ~3 $s^{-1}$ in the OFR case shown in Fig. 3, while
estimated to be only up to ~0.1 $s^{-1}$ in the ambient case, given the $RO_2$ concentration in the simulation.

It is known that $RO_2$ + OH can be a significant $RO_2$ loss pathway in OFR, especially when OH and

$HO_2$ production is relatively strong (higher relative humidity (RH) and UV). We have previously advised
to avoid such conditions in low-NO chemistry based on simplified modeling, because of its high-NO-like
organic product (Peng et al., 2019). Here our chemically explicit modeling results show that the
limitation for OFR chemistry caused by $RO_2$ + OH may not be as serious as suggested by Peng et al.
(2019), at least in terms of $OHR_{VOC}$ and, to some extent, of organic composition (Fig. 3). The condition
of the OFR case shown in Fig. 3 (30% RH, medium UV, and 10 $s^{-1}$ initial OHR) is a compromise between
the goals of reaching an equivalent photochemical age of ~10 d, avoiding significant non-tropospheric
organic photolysis, and keeping a more atmospherically relevant $RO_2$ chemistry (Peng et al., 2016; Peng
et al., 2019; Peng and Jimenez, 2020). In this case, the fractional contribution of $RO_2$ + OH to $RO_2$ fate
is still sizable (>30%). However, the evolution of the composition of monofunctional species in this OFR
case before the $OHR_{VOC}$ peak equivalent age is similar to that in the ambient case (Fig. 3), as
hydroperoxide production through $RO_2$ + $HO_2$ is still the main loss pathway of the first-generation $RO_2$
and RO produced from $RO_2$ + OH can also form ketones, i.e., the main second-generation products. The
other main fate of RO, i.e., isomerization, leads to slightly faster production of multifunctional species,
since the product of the recombination of the immediate product of this isomerization, i.e., an alkyl
radical, with $O_2$ is already a bifunctional $RO_2$. This isomerization also creates a hydroxyl group on the C
backbone, resulting in a relatively high share of hydroxyl in the functional groups of the multifunctional
species (Fig. 4).

Before the $OHR_{VOC}$ peak, as $OH_{exp}$ increases, carbonyls accumulate. They are prone to Norrish-

type photochemistry (Turro et al., 2009) which plays a major role in the OHR evolution after the $OHR_{VOC}$
peak in the ambient cases by breaking C10 species into smaller molecules (Fig. 3). Alkenes, which can
only be produced via Norrish Type II reaction in this case (Turro et al., 2009), are non-negligible OHR
contributors around $3 \times 10^{11}$ molecules $cm^{-3}$ s. A set of oxygenated C1 and C2 species are also largely
produced through organic photolysis followed by reactions with ($O_2$ and) $HO_2$. Organic photolysis, along
with OH reaction pathways, can also produce $RO_2$. Self- and cross-reactions of acylperoxy radicals,
which are formed in significant amounts in this $OH_{exp}$ range, can rapidly generate alkoxy radicals
(Orlando and Tyndall, 2012), which may decompose subsequently (Ziemann and Atkinson, 2012). At
$\sim 1 \times 10^{12}$ molecules $cm^{-3}$ s, the small species produced account for about half of $OHR_{VOC}$ in the ambient
cases (Fig. 3). In the OFR cases, organic photolysis is usually much weaker than in the atmosphere (Peng
et al., 2016; Peng and Jimenez, 2020), as the negligible OHR of alkenes in the OFR case in Fig. 3 also
indicates. However, fragmentation of multifunctional species does not appear to be significantly
weaker in the OFR case than in the ambient case shown in Fig. 3. This is largely due to fast $RO_2$ + OH.
The reactions of acylperoxys with OH lead to direct fragmentation (Orlando and Tyndall, 2012). Highly
functionalized RO can also form from the reactions of multifunctional $RO_2$ with OH, and then often
rapidly decompose. $RO_2$ + OH also results in a major difference of the OFR case from the ambient case
at high equivalent ages (Fig. 3), i.e., lower (higher) OHR contribution from $CH_3OOH$ (HCHO) in OFR than
in the atmosphere. Most $CH_3OO$ reacts with OH to produce $CH_3O$ then HCHO in the OFR case, leaving a
minor fraction of $CH_3OO$ reacting with $HO_2$ to form $CH_3OOH$.
**3.2.2   Chamber cases**

Three types of chamber simulations, without gas-particle-wall partitioning, with gas-particle

partitioning only (no wall), and with gas-particle-wall partitioning, are considered in this study. The first
type has almost the same results as the ambient cases in terms of the evolution of OHRVOC and its
composition as a function of $OH_{exp}$ (Fig. 1). Despite different spectra, sunlight and chamber lights cover
the same wavelength range and usually generate oxidizing agent radicals (e.g., OH and $HO_2$) in similar
amounts. Therefore, all key parameters are very similar between the ambient case with constant solar
radiation and the chamber case (with 10 $s^{-1}$ initial OHR) and without gas-particle-wall partitioning. This
explains the high similarities between the results of the ambient cases and the chamber cases without
wall partitioning. Unfortunately, the lack of wall partitioning is not realistic for current chambers.

The chamber cases with gas-particle-wall partitioning at lower initial OHR, which are realistic,

show very large deviations from the ambient cases that cannot be explained by gas-particle partitioning
only (Figs. 1, 3 and S4). These deviations are mostly due to wall partitioning of OVOCs around the $OH_{exp}$
of the OHR peak and at lower $OH_{exp}$. In this $OH_{exp}$ range, the wall removes about half of the OHR of
decane oxidation intermediates/products (Figs. 1 and 2) and hence also almost removes the OHRVOC
peak in the relevant chamber cases (Fig. 1). In detail, some decyl hydroperoxides partition to the wall
in the chamber case shown in Fig. 3, as decyl is a relatively large alkyl and leads to hydroperoxides of
sufficiently low volatility to promote wall partitioning. The C10 ketones, usually of higher volatility than
the corresponding hydroperoxides, do not show significant wall partitioning (Fig. 3), while about half of
the multifunctional C10 species, of even lower volatility than the corresponding hydroperoxides, are
partitioned to the wall.

At higher $OH_{exp}$ (>$2x10^{11}$ molecules $cm^{-3}$ s), more multifunctional species stay in the gas phase

or partition to the particle phase (Figs. 3 and S4). Those in the gas phase are formed via C10
fragmentation and are thus of higher volatility (Fig. 3). Those partitioned to the particle phase instead
of the wall are due to a higher organic aerosol concentration resulting from accumulation during a long
oxidation. The substantial partitioning of multifunctional species to aerosol and the wall also slows
down their oxidative evolution in the gas phase relative to the ambient cases (Fig. 4). The degree of
oxidation of products partitioned to the particle phase increases since low $OH_{exp}$. This is due to a
volatility fractionation caused by the lower ability of the particle phase to absorb condensable organic
gases than that of the wall phase at this $OH_{exp}$. At low organic aerosol loading, the relative potential of
particle partitioning to wall partitioning for organic gases of higher functionalization is higher than for
those of lower functionalization.

At higher initial OHR (100 s-1), and hence higher organic aerosol loading, condensable gases

have a significantly higher tendency of partitioning to the particle phase. The reduction of OHR of the
higher initial OHR chamber case with aerosol partitioning only (no wall partitioning) relative to the
purely gas-phase case is comparable to the lower initial OHR case with gas-particle-wall partitioning at
low $OH_{exp}$ (before the OHR peak) (Fig. S4). At high $OH_{exp}$, this OHR reduction is even stronger than in
the lower initial OHR case with gas-particle-wall partitioning, as partitioning of OVOCs to the particle
phase dominates over that to the walls.

As $OH_{exp}$ increases and large multifunctional species are formed in increasing amounts from

oxidation, their substantial partitioning to aerosol and the wall decreases the OHR of decane oxidation
intermediates/products by a factor up to 8 around 1x1012 molecules cm-3 s compared to the chamber
cases without gas-particle-wall partitioning (Figs. 2 and S4). At higher OHexp (long oxidation times) gas-
phase concentrations of partitioning species decline, allowing reverse partitioning back from the
particle phase and the wall which then serves as a source rather than a sink. As a result, the ratio of the
OHR of oxidation intermediates/products in the chamber case with gas-particle-wall partitioning to that
without this partitioning decreases (Fig. 2).
**3.2.3  OH and $HO_x$ recycling ratios**

As discussed above, we also compute OH ($\beta_1$) and $HO_x$ ($\beta_2$) recycling ratios in decane oxidation.

Note that these quantities also include OH and $HO_2$ generated as a result of organic photolysis. The
differences in these recycling ratios between the simulated cases are relatively small. $\beta_1$ is close to 0 at
$OH_{exp}$ < ~$1x10^{10}$ molecules $cm^{-3}$ s (Fig. 1), as the initial reaction of decane with OH only produces an $RO_2$
and subsequently C10 hydroperoxides, and no $HO_x$. Then $\beta_1$ undergoes a fast increase between ~$1x10^{10}$
and $1x10^{11}$ molecules $cm^{-3}$ s (Fig. 1), as the further oxidation of C10 hydroperoxides to ketones fully
recycles OH ($R_1$-CH(OOH)-$R_2$ + OH $\rightarrow$ $R_1$-CO-$R_2$ + $H_2O$ + OH) in the ambient and chamber cases.
Nevertheless, $\beta_1$ only increases up to ~0.4 at this stage in the ambient and chamber cases, as oxidation
of C10 hydroperoxides to dihydroperoxy species and precursor oxidation also account for a substantial
fraction of OH loss but do not recycle it. In the OFR cases, $\beta_1$ only increases up to ~0.2–0.3 at this stage,
since $RO_2$ + OH starts to be active but does not recycle OH. Then, $\beta_1$ roughly plateaus up to ~$1 \times 10^{12}$
molecules $cm^{-3}$ s, as the overall effect of the decrease in hydroperoxy concentration, reducing OH
recycling, and the increase in the concentration of acylperoxy, enhancing OH recycling through its
reaction with $HO_2$ (Orlando and Tyndall, 2012), is relatively small. Finally, $\beta_1$ gradually decreases to 0
(Fig. 1), as all OVOCs degrade to highly oxidized C1 species, i.e., HCHO, HCOOH, CO, which only have
$HO_2$ recycling but no OH recycling, and the unreactive $CO_2$.
The $HO_x$ recycling ratio ($\beta_2$) in decane oxidation is similar to $\beta_1$ before ~$1 \times 10^{11}$ molecules $cm^{-3}$ s
for the ambient and chamber cases, as only OH (but not $HO_2$) is recycled at this stage. $\beta_2$ is a little higher
in the OFR cases than in the other cases at this stage because of the $HO_2$ recycling by $RO_2$ + OH.
However, at higher $OH_{exp}$, $\beta_2$ continues to increase with $OH_{exp}$ to a final value of 1 (Fig. 1). This difference
between $\beta_1$ and $\beta_2$ is by definition due to $HO_2$ recycling. Its significance rises in parallel with that of
organic photolysis, which can often produce HCO radicals and acylperoxy radicals. The former
extremely rapidly undergoes HCO + $O_2 \rightarrow$ CO + $HO_2$; the latter can rapidly convert peroxy radicals to
alkoxy radicals (Orlando and Tyndall, 2012), which may then react with $O_2$ to generate $HO_2$ (Ziemann
and Atkinson, 2012). At very high $OH_{exp}$ ($10^{12} - 10^{13}$ molecules $cm^{-3}$ s), reactive highly oxidized small
VOCs are the dominant OHR contributors and many of them recycle $HO_2$ during their oxidation by OH
(Fig. 3). Finally,  once CO becomes the only remaining OHR contributor, $\beta_2$ is 1.
**3.3    m-Xylene**
Most features in m-xylene oxidation can be explained based on similar discussions as for decane
oxidation in Section 3.2. $OHR_{VOC}$ also has a maximum during the oxidation (Figs. 1 and S4), as most of
the direct products of m-xylene oxidation by OH, i.e., the unsaturated carbonyl (MXYEPOXMUC in MCM
v 3.2 notation, see Scheme S1), the unsaturated endo-cyclic peroxide (MXYBPEROOH), and xylenols,
are more reactive toward OH than m-xylene. The OHR of these initial products is enhanced much more
during the oxidation of m-xylene than of decane, owing to the creation of C=C bonds in many post-
aromatic (ring-opening) products, hence the $OHR_{VOC}$ peak enhancement in m-xylene oxidation is larger
than in decane oxidation. Because the reaction rate constant of m-xylene with OH slightly exceeds that
of decane, the $OHR_{VOC}$ peak in m-xylene oxidation occurs at slightly lower $OH_{exp}$ than in decane
oxidation (Fig. 1). In the OFR case under the same condition as the decane case shown in Fig. 3, the
evolution of OHR of the stable organic species is again similar to that in the ambient case. And $OHR_{VOC}$
is higher in the OFR case again mainly due to OHR from $RO_2$ (Fig. 1 and S4). Several main first- and
second-generation products are already highly functionalized through fast $O_2$ addition (Scheme S1) and
they are also often unsaturated and prone to further functionalization. Therefore, the degree of
functionalization in saturated aliphatic multifunctional species is much higher in m-xylene than in
decane oxidation (Fig. 4). Also, as several aromatic-scheme-specific reaction types occur in the early
stages of m-xylene oxidation, e.g., endo $O_2$ addition (creating -OO- etc.) and ring-opening (creating -
CO-, -CHO etc.), multifunctional species functionality is more diverse than in decane oxidation (Fig. 4).
Photolysis again plays a role in species fragmentation and the production of highly oxidized C1 and C2
species after the OHR$_{VOC}$ peak (Fig. S4).

At low OH$_{exp}$ and that of the OHRVOC peak, particle and wall partitioning also substantially
reduces the OHRVOC in the relevant chamber cases of m-xylene oxidation while the OHRVOC reduction
due to partitioning to the particle phase is smaller than that due to the wall (Figs. 1, 2, S4 and S5). The
precursor (m-xylene) is a C8 species and even many first-generation products of its oxidation are highly
oxygenated (Scheme S1) lower-volatility species. The relative reduction of OHR of the
intermediates/products also increases with OH$_{exp}$ before the OH$_{exp}$ of the OHR peak, as volatile species
are oxidized and become more prone to wall partitioning (Figs. 2 and S4). At higher OH$_{exp}$, more
condensed organics are partitioned to the particle phase because of high organic aerosol concentration
(Fig. S4) and the wall and aerosol again serve as OVOC source (Fig. 2).

The evolution of $\beta_1$ and $\beta_2$ in m-xylene oxidation is somewhat different than in decane oxidation
(Fig. 1). In the ambient cases, they are non-negligible even at OH$_{exp}$ as low as $1\times10^9$ molecules cm$^{-3}$ s
($\sim$0.05 and $\sim$0.45, respectively). OH is mainly recycled from one of endo-cyclic peroxide routes (m-
xylene + OH + 2O$_2$ $\rightarrow$ MXYBIPERO2; MXYBIPERO2 + HO$_2$ $\rightarrow$ MXYBPEROOH + O$_2$; MXYBPEROOH + OH $\rightarrow$
MXYOBPEROH + H$_2$O + OH (Scheme S1)), which involve various functional groups and open the
possibility of radical recycling. The third step of this route is very fast (with a rate constant on the order
of 10$^{-10}$ cm$^3$ molecule$^{-1}$ s$^{-1}$). Once the second step takes place, the third step contributes to OH recycling.
However, in the OFR cases with strong water vapor photolysis (not in the other OFR cases), the third
step does not play a significant role and $\beta_1$ is $\sim$0 at very low OH$_{exp}$ (Fig. 1) due to the relatively slow
second step (RO$_2$ + HO$_2$). In the former cases, this is due to the relatively slow second step (RO$_2$ + HO$_2$),
while in the latter cases, the highly oxygenated compounds partition to the wall even more rapidly (in
hundreds of s; Krechmer et al., 2016) than their reactions with HO$_x$. Strong HO$_2$ recycling occurs in all
simulated cases from the beginning of the oxidation (Fig. 1), since two of the three major channels of
m-xylene + OH (i.e., those forming MXYEPOXMUC and xylenol, respectively) produce HO$_2$ as well.

As more multifunctional species are formed (particularly through ring-opening) near the OH$_{exp}$
of the peak OHR$_{VOC}$, HO$_x$ recycling is also active, with $\beta_1$ increasing and $\beta_2$ remaining high (Fig. 1). There
is a high peak in $\beta_2$ for the chamber case with high initial OHR (100 s$^{-1}$) and no aerosol or wall
partitioning. It results from RO$_2$ cross-reactions, many of which produce alkoxy radicals that
subsequently yield carbonyls and HO$_2$ through reactions with O$_2$ (Orlando and Tyndall, 2012). RO$_2$ cross-
reactions are significant in that OH$_{exp}$ range also because i) high precursor concentration translates into
higher RO2 concentration and ii) acylperoxy radicals, whose reactions with other RO$_2$ are fast (Orlando
and Tyndall, 2012), are rapidly formed from the oxidation of -CHO groups in the ring-opening products
(Scheme S1). The chamber case with high initial OHR and gas-particle-wall partitioning does not have
such a high $\beta_2$ peak, because of fast partitioning of the oxidation products containing -CHO groups to
the aerosol and wall phases, which significantly reduces acylperoxy radical concentration around the
OH$_{exp}$ of the peak OHRVOC. At higher OH$_{exp}$, calculated $\beta_1$ and $\beta_2$ become less reliable, since remaining
apparent OHR contributors may in fact be persistent artifacts of the incompleteness of the (hand-
written) m-xylene oxidation mechanism which may substantially bias $\beta_1$ and $\beta_2$ when the

concentrations of remaining OHR contributors should be generally low. Therefore, we do not try to interpret the features in $\beta_1$ and $\beta_2$ at high $OH_{exp}$ for m-xylene oxidation.

**3.4    Isoprene**

The most salient difference of the $OHR_{VOC}$ evolution in the photooxidation of isoprene from that of the other precursors in this study is the lack of $OHR_{VOC}$ peak in the isoprene cases (Figs. 1 and S5). The decrease in $OHR_{VOC}$ all along this photooxidation is expected since the reaction of isoprene with OH is very fast (at $1 \times 10^{-10}$ $cm^3$ molecule$^{-1}$ s$^{-1}$; Atkinson and Arey, 2003) and all intermediates/products of this photooxidation react with OH more slowly than isoprene. The $OHR_{VOC}$ of the intermediates/products peaks slightly after an $OH_{exp}$ of $1 \times 10^{10}$ molecules $cm^{-3}$ s (Fig. 1). At this $OH_{exp}$, the main type of the first-generation products, oxygenated unsaturated species (e.g., isoprene-derived unsaturated hydroxyl hydroperoxides (ISOPOOH)), are largely produced from isoprene + OH and their loss rates (with rate constant with OH slightly lower than that of isoprene) reach the maxima (Fig. S5). Further oxidation leads to the loss of all C=C bonds in the isoprene C backbone and thus a substantial drop of the OHR of the molecule.

Before $OH_{exp}$~$5 \times 10^{10}$ molecules $cm^{-3}$ s in isoprene photooxidation, the main deviations from the ambient cases shown by the chamber cases with wall partitioning are again caused by wall partitioning of multifunctional species, but their relative magnitudes are different than in the photooxidations of decane and m-xylene, with the impacts of wall partitioning being smaller (Fig. S5). Oxygenated species derived from isoprene, a C5 species, should be generally more volatile and less prone to wall partitioning than those derived from decane and m-xylene. On the other hand, isoprene reacts with OH much more rapidly than do decane or m-xylene, creating a larger deviation from the steady state for $RO_2$ directly derived from isoprene and a more remarkable decrease in the OHR of the first-generation products (Fig. S5). In the OFR case shown in Fig. S5, $RO_2$ contributes negligibly to $OHR_{VOC}$, since many first-generation isoprene-derived $RO_2$ have other very fast loss pathways and the very fast decay of isoprene cannot sustain $RO_2$ production at $OH_{exp}$>~$1 \times 10^{10}$ molecules $cm^{-3}$ s.

After $OH_{exp}$~$5 \times 10^{10}$ molecules cm-3 s, the deviation caused by chamber wall partitioning becomes more significant as highly oxidized and lower-volatility multifunctional species (Fig. 4) are formed in significant amounts (Figs. 2 and S6). However, aerosol partitioning does not become more significant as in the chamber cases of isoprene oxidation aerosol formation is always so small that partitioning to the particle phase never competes with that to the walls. At very high $OH_{exp}$, the wall again acts as a source of OVOCs in isoprene oxidation, as in those of the other precursors (Fig. 2). The deviations of OFR cases from the ambient cases are mainly caused by $RO_2$ + OH and lack of organic photolysis. These two effects lead to too much HCHO produced and inefficient production of other C1 and C2 species (Fig. S5).

To test whether one of the issues, i.e., lack of organic photolysis in OFR, can be mitigated by adding tropospherically-relevant UV sources, we perform two additional simulations. Adding the emissions corresponding to high Hg lamp setting with five times the UV of the CU Chamber (a rough upper limit for experimental implementation) has negligible effect (Fig. S6). To reach a ratio between

tropospherically-relevant UV (UVA+UVB) intensity and OH concentration similar to that in the ambient
case with constant sunlight requires addition of a chamber light ~10000 times stronger than the CU
Chamber light. Such a strong UV source is obviously not realistic, and, while it does increase both early
organic photolysis and the relative contribution of C1 and C2 photoproducts to $OHR_{VOC}$ around $2x10^{11}$
molecules $cm^{-3}$ s (Fig. S6), it increases the deviation of this OFR case from the ambient cases at very
high $OH_{exp}$, where oxidation of C1 and C2 species to CO proceeds much more rapidly than in the
atmosphere.

Product functionality in isoprene oxidation is more diverse than in decane oxidation (Fig. 4). This

is due to both the propensity of the isoprene C=C bonds to addition of various groups, and the active
isomerization of isoprene oxidation intermediates (Wennberg et al., 2018). Notably, epoxy groups in
species such as isoprene-derived epoxydiol (IEPOX) account for a large fraction of saturated product
functionality (Fig. 4), particularly at $OH_{exp}$ on the order of $10^{10}$ molecules $cm^{-3}$ s. In the gas phase of the
chamber cases with wall partitioning, the overwhelming majority of saturated multifunctional organic
molecules are IEPOX up to $1x10^{11}$ molecules $cm^{-3}$ s (Fig. 4), as more highly-oxidized species mostly
partition to the wall.

IEPOX formation from isoprene-derived hydroperoxide (ISOPOOH) oxidation by OH (ISOPOOH +

OH → IEPOX + OH) leads to the peak of OH recycling around $3x10^{10}$ molecules $cm^{-3}$ s (Fig. 1). OH
recycling is active even at very low $OH_{exp}$ ($1x10^9$ molecules $cm^{-3}$ s) because a significant amount of
ISOPOOH forms early and can recycle OH through its oxidation, except in the OFR cases with strong
water vapor photolysis, where ISOPOOH cannot be efficiently formed from first-generation $RO_2$. $HO_2$
recycling is also active in the entire course of the photooxidation (Fig. 1), because of a number of
isomerization and photolysis pathways that form alkoxy radicals and highly oxidized C1 species such as
HCOOH, HCHO, and CO at very high $OH_{exp}$ (Fig. S5).
**3.5    Trends in OHR per C atom**

To explore some general trends of OHR evolution in VOC photooxidation, simulations are

performed for the ambient cases with constant UV for two additional alkanes between methane and
decane, i.e., butane and heptane. The results of these simulations are compared to the existing
analogous cases in Fig. 5. For all cases, the $OHR_{VOC}$ peak height decreases and the $OH_{exp}$ of the $OHR_{VOC}$
peak shifts towards lower $OH_{exp}$, as the C number of the precursor alkane increases. This can be
explained by the fact that the OH rate constants of these alkanes increase with C number, and suggests
a possible general trend between OHR peak location and C number.

To explore these trends further, we calculate the OHR per unit starting concentration of C atom

(in the precursor) in all ambient cases with constant UV in this study (Fig. 5b). In this study, $CO_2$ is not
included initially but produced during the oxidation. Therefore, C atoms in the produced $CO_2$ are taken
into account in the calculation of OHR per C atom. For real atmospheric cases, initial $CO_2$ is present but
should not be considered in this calculation. Note that OHR per C atom has a unit of $cm^3$ $atom^{-1}$ $s^{-1}$ and
represents the average contribution to the rate constant with OH of all considered C atoms. Despite
large differences among the reactivities of these precursors, the OHR per C atom in the simulations of
all precursors but methane converges near an $OH_{exp}$ of $3 \times 10^{11}$ molecules $cm^{-3}$ s, and then follows a very
similar downward trend (Fig. 5b). This $OH_{exp}$ value is roughly where saturated multifunctional species
have their maximal relative contribution to the $OHR_{VOC}$ (Figs. 3 and S5). Even in the ambient cases of m-
xylene oxidation, saturated multifunctional species also account for about half of $OHR_{VOC}$ when the
contribution of aromatics, some of which may artificially persist due to mechanism incompleteness, is
excluded (Fig. S4). Also, at $OH_{exp} > {\sim}3 \times 10^{11}$ molecules $cm^{-3}$ s, a C atom in saturated multifunctional
species on average has at least 0.3 functional groups in the ambient cases (Fig. 4), and the functional
group composition is relatively diverse at this $OH_{exp}$. Therefore, the convergence value of OHR per C
atom of ${\sim}2 \times 10^{-12}$ $cm^3$ $atom^{-1}$ $s^{-1}$ at ${\sim}3 \times 10^{11}$ molecules $cm^{-3}$ s can be largely regarded as a relatively
invariant average of those of secondary H and α-H of various O-containing functional groups. Note that
this average is for NO-free conditions and can be lower at high NO due to deactivating effects of N-
containing groups formed during oxidation (Isaacman-VanWertz and Aumont, 2021).
Before the convergence, isoprene has the highest OHR per C atom (on the order of $10^{-11}$ $cm^3$
$atom^{-1}$ $s^{-1}$) among the precursors and intermediates/products (Fig. 5b), because of its conjugated C=C
bonds. The OHR per C atom of its first-generation oxidation products is slightly lower and close to that
of the oxidation intermediates/products of m-xylene, as the main contributors in both cases are
oxygenated monoalkenes. The average OHR per C atom of the studied alkanes increases with C number
(Fig. 5b), with the upper limit around $1 \times 10^{-12}$ $cm^3$ $atom^{-1}$ $s^{-1}$ consistent with Kwok and Atkinson (1995),
since the less-reactive -$CH_3$ groups (with OHR per C atom of ${\sim}1 \times 10^{-13}$ $cm^3$ $atom^{-1}$ $s^{-1}$) contribute
proportionally less to molecular OHR as C number increases. Conversely, the early-stage products of
alkane oxidation (mainly alkyl monohydroperoxides) show higher average OHR per C atom for shorter
molecules (Fig. 5b), owing to the activating (increasing OHR) contribution of the -OOH group.
Following the convergence of OHR per C atom, this quantity in all non-methane ambient cases
in this study sees a similar decay (Fig. 5b). This coincides with multifunctional species broken into small
highly oxidized C1 and C2 compounds. Although among them there are species with OHR per C atom >
$5 \times 10^{-12}$ $cm^3$ $atom^{-1}$ $s^{-1}$ (e.g., $CH_3OOH$, $CH_3CHO$, and HCHO), the average OHR per C atom of these C1 and
C2 species are mainly governed by those reacting more slowly (e.g., HCOOH and particularly CO) and
hence reaching higher concentrations amid the fast decay of multifunctional species. The similar fast
drop of OHR per C atom after $OH_{exp} {\sim} 1 \times 10^{12}$ molecules $cm^{-3}$ s for various precursors implies a transition
from OHR from saturated multifunctional molecules to OHR from CO before the final oxidation to $CO_2$
(which has zero OHR).
**3.6  Total OH consumption for each precursor**
Integrating OHR per C atom over $OH_{exp}$ allows us to assess the average number of OH molecules
consumed by each C atom during the entire course of oxidation. This quantity can also be apportioned
to the contributions of different OH reactants (Fig. 6). Due to incomplete oxidation of several species,
especially CO, the value of this quantity for an oxidation with all C atoms ending up with $CO_2$ should be
higher than those at simulation end ($OH_{exp} {\sim} 4 \times 10^{12}$ molecules $cm^{-3}$ s). We correct this in Fig. 6 by
including additional contribution of CO to make its total contribution 1, since CO, the typical
penultimate product, consumes one OH molecule in its final oxidation, but is still present in significant
quantities at the end of our simulations. Thus, each C atom reacts with OH ~3 times in the course of the
oxidation of isoprene and decane to $CO_2$ (Fig. 6). A simplistic and chemically intuitive explanation for
this number is that the average oxidation state ($\overline{OS_C}$) of both isoprene and decane C atoms is ~-2, and
needs to increase to the value of +4 in $CO_2$ at the end of the oxidation. A C1 unit reacting once with OH
likely increases its $\overline{OS_C}$ by ~2. This increase is usually realized by an abstraction of H atom by OH or an
addition of OH ($\overline{OS_C}$+ 1), followed by an abstraction of H atom by $O_2$ or an addition of $O_2$ ($\overline{OS_C}$+ 1). Note
that ~3 OH consumed per C atom oxidized to $CO_2$ is likely an upper limit, since the mechanisms in this
study do not include $RO_2$ autoxidation (Crounse et al., 2013; Ehn et al., 2014), which reduces the
number of OH needed for complete VOC oxidation. Also, in a real low-NO environment, NO is still
present in low concentrations and converts $RO_2$ to RO. RO may undergo H abstraction through
isomerization or reaction with $O_2$, which also lowers the number of OH needed, although the effect is
usually small. The number of OH consumed per C atom in m-xylene oxidation is slightly lower than 3
(Fig. 6) because of the multiple addition of $O_2$ following a single OH addition in the initiation reaction,
i.e., m-xylene + OH.
**4    Summary and conclusions**
Using the fully explicit GECKO-A model, we simulated OHR evolution in the photooxidation of
several types of VOCs (i.e., alkane, alkene, and aromatic) without NO until very high equivalent
photochemical ages (>10 d) under a variety of conditions (in the atmosphere, chamber, and OFR). We
analyzed the simulations in detail and found a number of common features as well as some differences
resulting from certain precursors. These features are summarized below:
-    All simulated non-methane cases very roughly follow this general oxidation chain pattern: precursor
→ first-generation products → (second-generation products →) multifunctional species → highly
oxidized C1 and C2 species → CO (or HCOOH) → $CO_2$. These species are generally not at steady
state and gain significance/predominance one after another in the entire course of oxidation.
Simulation results suggest that fragmentation products are not formed in significant amounts until
the late stage of the oxidation, which would be a key difference from studies of OHR evolution in
high-NO VOC oxidation (Nakashima et al., 2012; Sato et al., 2017).
-    In methane oxidation, the intermediates do not gain dominance in sequence. Instead, they
simultaneously increase as the oxidation proceeds, then simultaneously decrease when the
methane decay becomes significant. The OHR evolution in methane oxidation is close to the
idealized steady-state chain model, as the reaction of methane with OH is orders of magnitude
slower than those of its oxidation intermediates, which allows the intermediates to reach their
steady state.
The following discussion refers to the non-methane cases.
-    Where different types of species dominate $OHR_{VOC}$ in sequence, $OHR_{VOC}$ increases after the current
dominant type converts to one with a higher average OHR per C atom, and vice versa.
Photooxidations of alkanes and aromatics follow the increasing trend from precursor to saturated
multifunctional species (via alkyl monohydroperoxides) and from precursor to unsaturated
oxygenated species, respectively. The increase in aromatic oxidation is likely to be more significant,
since unsaturated oxygenated species are more reactive than saturated multifunctional species.
The conversions from multifunctional species to $CO_2$ lead to a decay of $OHR_{VOC}$ in both alkane and
aromatic photooxidations. $OHR_{VOC}$ in alkene photooxidation is likely to always drop rapidly during
C=C bond oxidation and more slowly afterwards.
- A relatively weak enhancement of OHR per C atom of a C atom with -OOH substitution can explain
the large range spanned by the precursors and their intermediates/products in this study at low
$OH_{exp}$. Around an $OH_{exp}$ of $3 \times 10^{11}$ molecules $cm^{-3}$ s, precursors are largely converted to saturated
multifunctional species (e.g., by addition to C=C bonds in unsaturated precursors and abstraction
of H atoms in saturated precursors), and the reactive mixtures of those precursors thus have similar
OHR per C atom. They then all follow the course: multifunctional species → highly oxidized C1 and
C2 species → CO (or HCOOH) → $CO_2$ and show similar decays of OHR per C atom.
- In decane and isoprene oxidation, our simulations show that each C atom consumes at most ~3 OH
molecules in the course of its oxidation to $CO_2$. This can be simplistically explained as 3 occurrences
of oxidation by OH that increase, by 2 each time, the $\overline{OS_C}$ of decane and isoprene (~-2) to that of
$CO_2$ (+4). The total number of OH consumed by each C atom is likely to be lower when $RO_2$
autoxidation can be included in the mechanism generation.
In general, the OHR evolution differences resulting from different precursors are larger than those due
to different conditions. The difference in $OHR_{VOC}$ between the ambient cases with constant and
diurnal sunlight is small. Nevertheless, physical conditions may still lead to significant differences,
which are summarized below:
- In current chambers, gas-wall partitioning can be a prominent issue that causes substantial wall
partitioning of certain OVOCs of lower volatility, depending on the chemical system under study.
The clearest example in this study is the substantial wall losses of C10 multifunctional species from
the gas phase in decane oxidation, and hence the remarkably lowered OHRVOC peak height in the
chamber simulation. The wall also preferentially absorbs more oxidized (and thus lower-volatility)
species, which alters the functional composition of gas-phase multifunctional species. The wall can
even serve as a source of multifunctional species at very high $OH_{exp}$, when the gas-phase
concentrations of those species are very low. The magnitude of the effects of wall partitioning also
depends on the size of the precursor, with the oxidation of larger precursors in chambers suffering
larger impacts of wall partitioning.
- The strong wall losses have important implications, as they can change our modeling results
substantially. Systematic OVOC gas-particle-wall partitioning corrections must be made for low-NO
oxidation chamber experiments that study $OHR_{VOC}$. In case of large precursors, highly chemically
explicit modeling will likely be necessary to infer the OHR of multifunctional species, which may
account for a large fraction of missing reactivity but suffer substantial wall losses. Although the few
existing chamber studies on $OHR_{VOC}$ evolution were all under high-NO conditions, which may result
in more fragmentation and higher-volatility products, the magnitude of wall partitioning of large
multifunctional species in this study is so substantial that we believe this effect would also be
important at high NO. Schwantes et al. (2017) considered wall partitioning in their modeling of o-
cresol oxidation based on MCM v3.3.1 but still could not achieve good agreement with the
measurements for a number of products. Considering this, one should not assume that it is
appropriate to neglect gas-particle-wall partitioning in high-NO chamber experiments, just based
on agreement between the high-NO chamber experiments and modeling with MCM-based
schemes and without gas-particle-wall partitioning corrections. Even for OHR studies with less
surface loss issues, e.g., ambient studies, a combination of gas-phase-only OHR measurement and
modeling may still not be adequate as reduction of OHR due to OVOC condensation on aerosols can
also be significant in some situations (Fig. S4). Therefore, condensed phases (particle and wall) need
to be included in future OHR studies to better assess the deviation of the actual OHR from a purely
gas-phase picture.
- OFR has two issues under certain conditions that can cause deviations from the ambient cases in
terms of $OHR_{VOC}$. Strong $RO_2$ + OH may significantly contribute to $OHR_{VOC}$. Interestingly, this type of
reaction does not seem to be able to substantially alter the composition of $OHR_{VOC}$. before the
$OHR_{VOC}$ peak. Besides, the conditions resulting in strong water vapor photolysis have already been
identified as those leading to atmospherically irrelevant $RO_2$ chemistry in low-NO OFR in previous
studies (Peng et al., 2019; Peng and Jimenez, 2020). As long as OFR users follow the guidelines for
experimental planning provided in those studies (and use a much lower UV lamp setting), strong
$RO_2$ + OH can be avoided, as shown in Fig. 1.
- The other main issue of OFR is lack of efficient organic photolysis, particularly at high $OH_{exp}$, when
multifunctional species break into highly oxidized C1 and C2 compounds. This problem has been
highlighted in previous studies (Peng et al., 2016; Peng and Jimenez, 2020) and been shown again
in the present work to be extremely difficult to avoid if a high $OH_{exp}$ is desired. However, the
conversion of multifunctional species into highly oxidized C1 and C2 species may not be much
slower in OFR than in the atmosphere, since $RO_2$ + OH, leading to RO formation and subsequently
its decomposition, may also play a major role in this conversion. This also results in significantly
higher (lower) production of HCHO ($CH_3OOH$) in OFR than in the atmosphere at high equivalent
ages.
With all the key findings in this study presented above, we believe that we have, to some extent,
addressed the issues of "missing reactivity", of model limitations, and of OHR in remote areas for OHR
studies raised by Williams and Brune (2015). With the fully explicit GECKO-A model, we speciated the
likely source of the "missing reactivity", i.e., multifunctional OVOCs. A contrast between the technical
issues in some isoprene and m-xylene simulations and the high consistency in the other cases highlights
the importance of the completeness of the mechanism (even beyond the MCM level) in OHR-related
modeling studies. Substantial wall partitioning of OVOCs in some chamber experiments highlights the
importance of better constraining "gas-to-surface deposition terms''. More studies, both modeling
(with highly chemically explicit mechanisms) and experimental (particularly low-NO), are needed to
achieve better model-experiment closure. Finally, this study may have opened up the possibility of
parameterizing the OHR evolution in (at least low-NO) VOC photooxidation as a function of $OH_{exp}$ only
with the often-available knowledge on the first- and second-generation products and the relevant SARs
such as Kwok and Atkinson (1995), as the OHR evolution beyond multifunctional species has been
shown to be similar for most VOC oxidations. This parametrization may be utilized in regional and global
models to better constrain OHR at high equivalent photochemical ages, e.g., in remote regions.

**Appendix A: The effect on OH reactivity of non-OH reactions in an OH-driven reaction chain**
1)      Consider the following reaction chain, where OH is the only oxidant:
$A_1 + OH \rightarrow A_2$,      $k_1$
$A_2 + OH \rightarrow A_3$,      $k_2$
$A_3 + OH \rightarrow A_4$,      $k_3$
…(etc.)
At steady state, $k_1[A_1][OH] = k_2[A_2][OH] = k_3[A_3][OH] =…= C$ (C is a constant).
Then OHR due to individual species, $OHR_i$, is equal to $C/[OH]$ and is identical for all species.
2)      Consider a parallel conversion of $A_1$ to $A_2$ by a means other than reaction with OH:
$A_1 + OH \rightarrow A_2$,      $k_1$
$A_1 + B \rightarrow A_2$,      $k_1'$, B ≠ OH
At steady state, $k_1[A_1][OH] + k_1'[A_1][B] = C$.
Therefore, $OHR_{A1} = k_1[A_1] < C/[OH]$.
3)      Now consider a reaction converting $A_1$ directly to $A_3$ occurring in parallel to reaction chain 1),
$A_1 + D \rightarrow A_3$,      $k_1''$, D ≠ OH
$A_1 + OH \rightarrow A_2$,      $k_1$
$A_2 + OH \rightarrow A_3$,      $k_2$
At steady state, $k_1''[A_1][D] + k_2[A_2][OH] = C$
And $OHR_{A1} < C/[OH]$, since $k_1[A_1][OH] = k_2[A_2][OH]$.

**Code/Data availability**
The chemical mechanisms generated and the outputs of the GECKO-A simulations in this study are
available upon request.

**Author contribution**
ZP and JLJ conceived the study. ZP designed the study. JL-T and ZP performed the simulations. HS, JL-T,
ZP, and JLJ developed the GECKO Loader and Plotter. ZP, JL-T, JJO, and BA made updates and
developments for GECKO-A. ZP, JL-T, JJO, BA, and JLJ analyzed the results. ZP led the manuscript writing
with inputs from all authors.

**Conflicts of interest**
There are no conflicts to declare.

**Acknowledgements**

This work was partially supported by NSF AGS-1822664 and AGS-1740610. We thank Sasha Madronich and Alma Hodzic for useful discussions. We would like to acknowledge high-performance computing support from Cheyenne (doi:10.5065/D6RX99HX) provided by NCAR's Computational and Information Systems Laboratory, sponsored by the National Science Foundation.

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

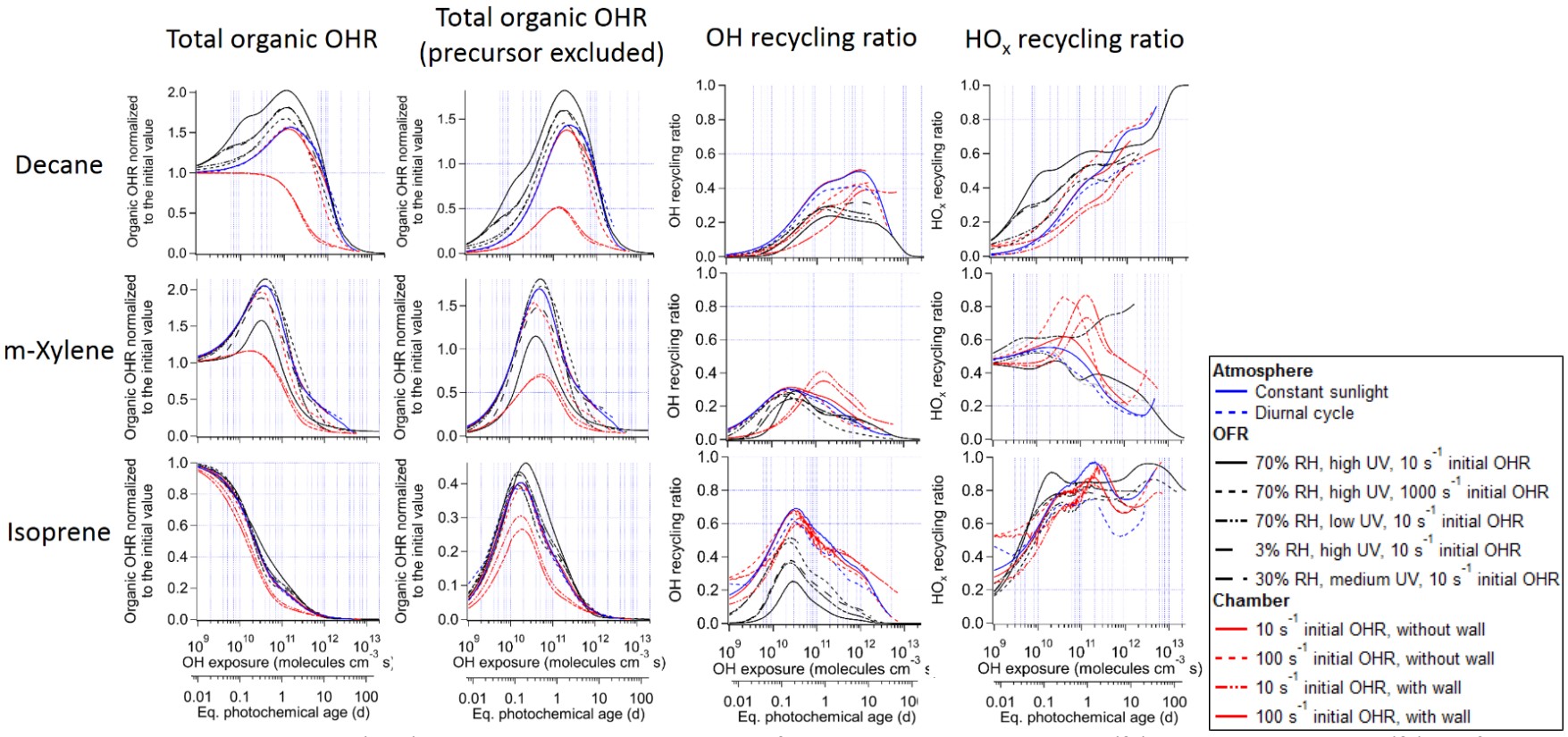

**Figure 1.** Total organic OH reactivity (OHR) with and without the contribution of the precursor, OH recycling ratio ($\beta_1$), and HO$_x$ recycling ratio ($\beta_2$) as a function of OH exposure (or equivalent photochemical age; second x-axis) in the photooxidations of decane, isoprene, and m-xylene under different conditions in the atmosphere, oxidation flow reactor (OFR), and chamber.

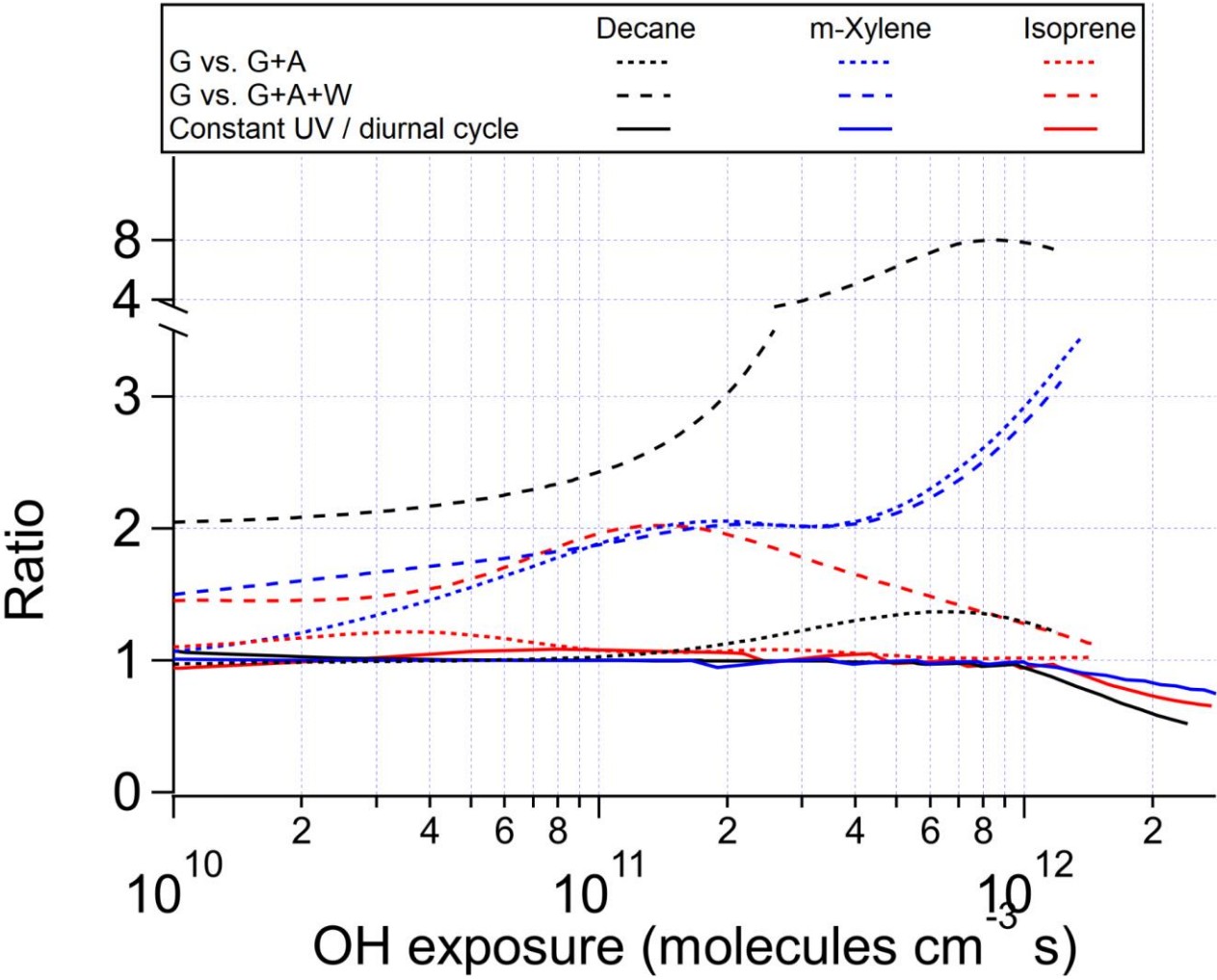

**Figure 2.** Ratios of OHR of the products present in the gas phase between the chamber cases without gas-particle-wall partitioning and i) with gas-particle (G vs. G+A) or ii) gas-particle-wall partitioning (G vs. G+A+W) at initial OHR of 10 s$^{-1}$, and between the ambient cases with constant and diurnal sunlight for the photooxidations of decane, m-xylene, and isoprene as a function of OH exposure.

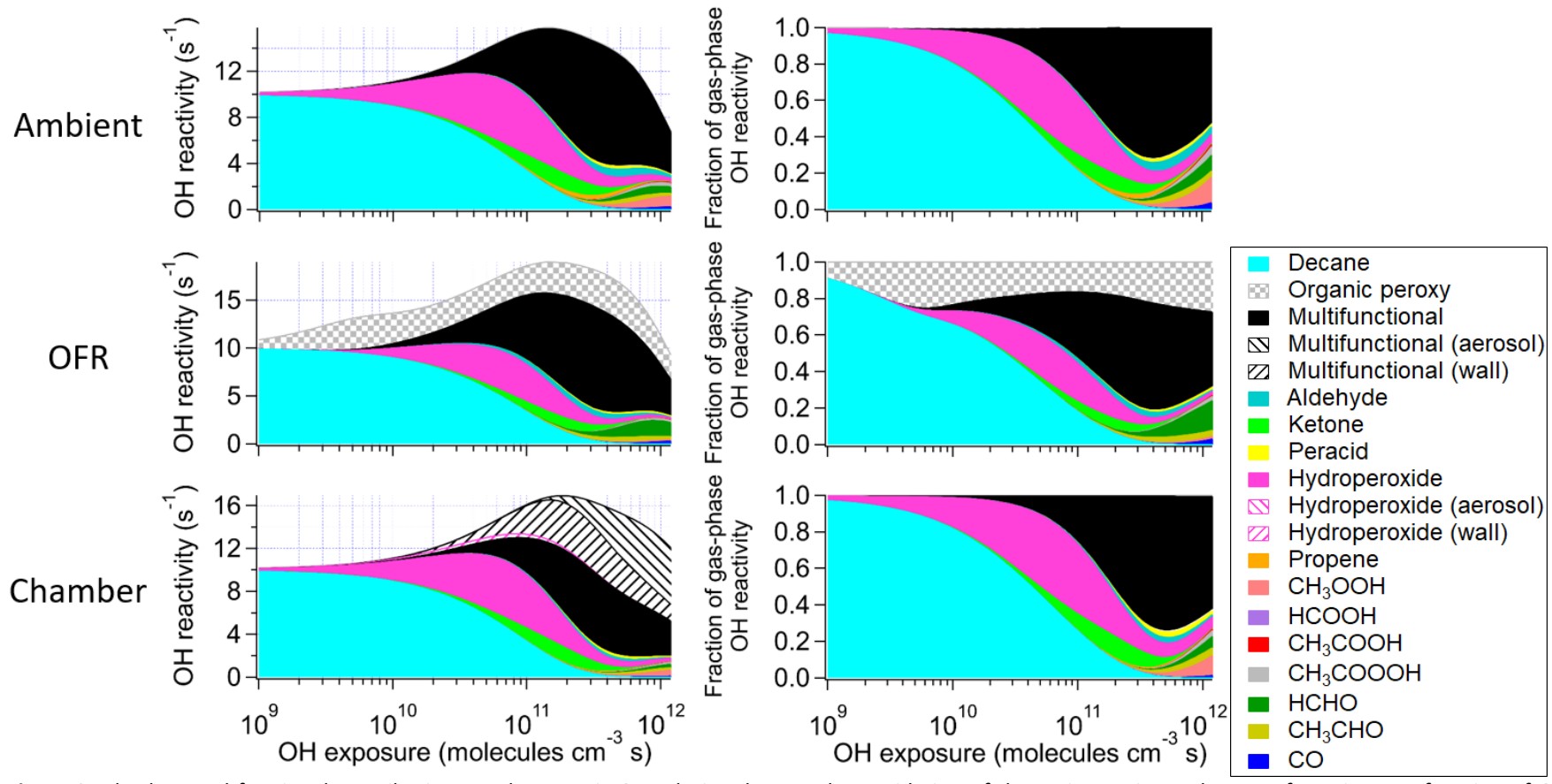

**Figure 3.** Absolute and fractional contributions to the organic OHR during decane photooxidation of the main species and types of species as a function of OH exposure in the ambient case with constant sunlight; the OFR case with relative humidity of 30%, medium UV lamp setting, and initial OHR of 10 s$^{-1}$; and the chamber case with initial OHR of 10 s$^{-1}$ and gas-wall partitioning. The types of species shown in this figure exclude the C1 and C2 species listed separately. The OHR of the particle- and wall-phase species are the values as if those species are gas-phase OHR contributors, although they actually do not react with OH in the simulations.

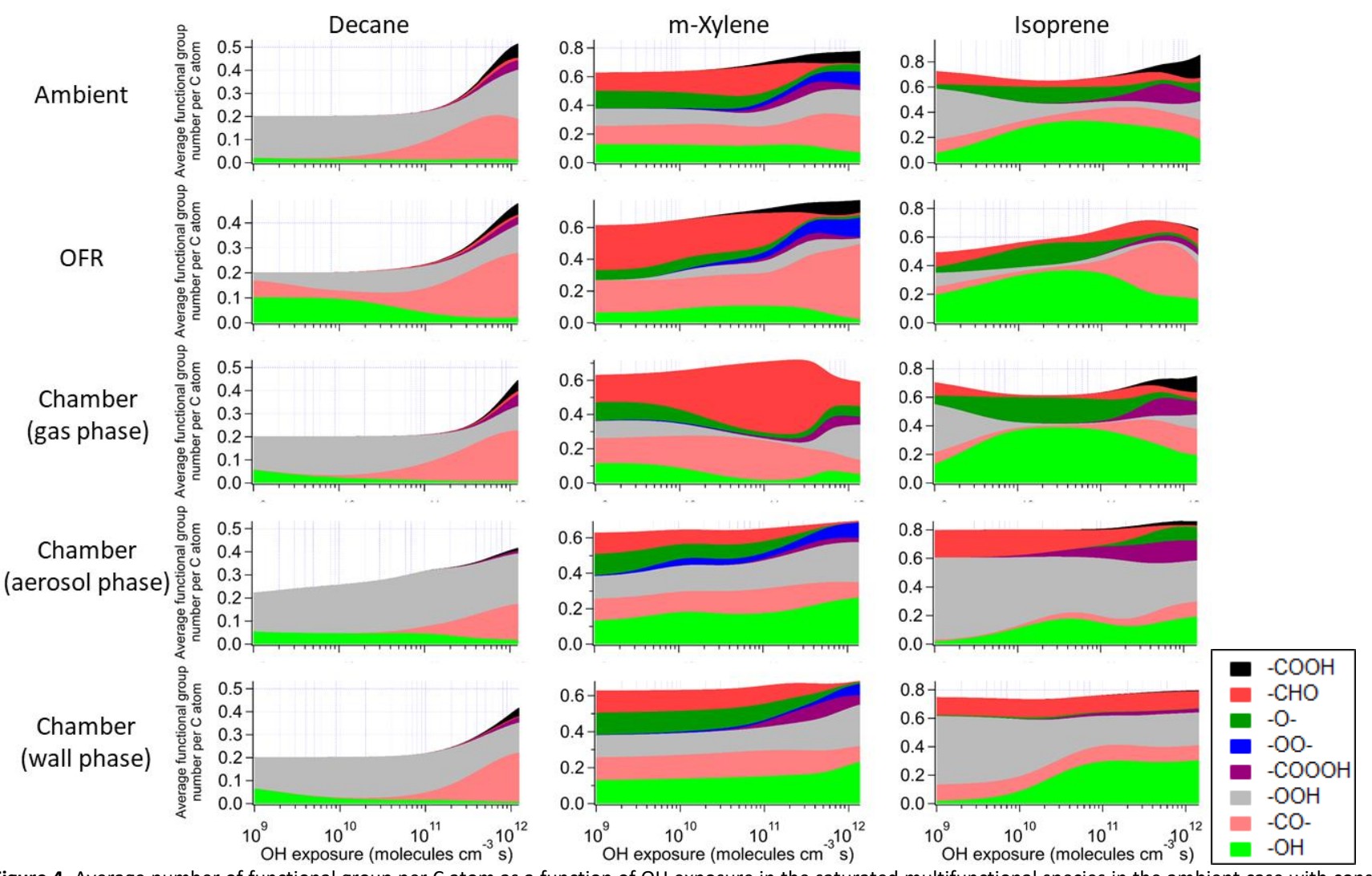

**Figure 4.** Average number of functional group per C atom as a function of OH exposure in the saturated multifunctional species in the ambient case with constant sunlight, the OFR case with relative humidity of 70%, high UV lamp setting, and initial OHR of 10 s$^{-1}$, and the gas, aerosol, and wall phases in the chamber case with initial OHR of 10 s-1 and gas-particle-wall partitioning of the photooxidations of decane, m-xylene, and isoprene. Note that the functional group "-O-" represents

ether, ester, and epoxy groups in the GECKO-A model.

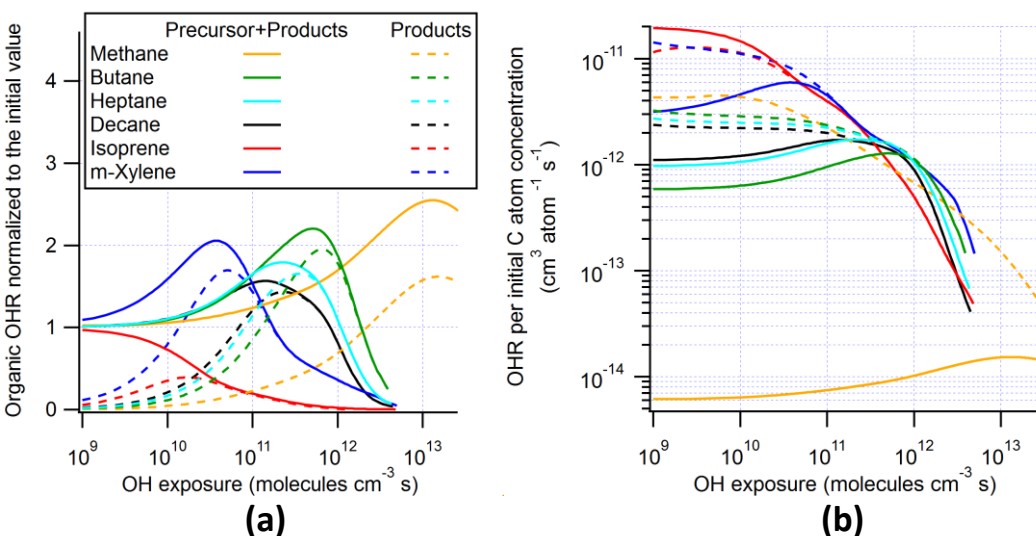

**Figure 5.** (a) OHR and (b) OHR per initial C atom concentration of the organics (including and excluding the precursor) as a function of OH exposure in the ambient cases with constant sunlight of the photooxidation of methane, butane, heptane, decane, isoprene, and m-xylene.

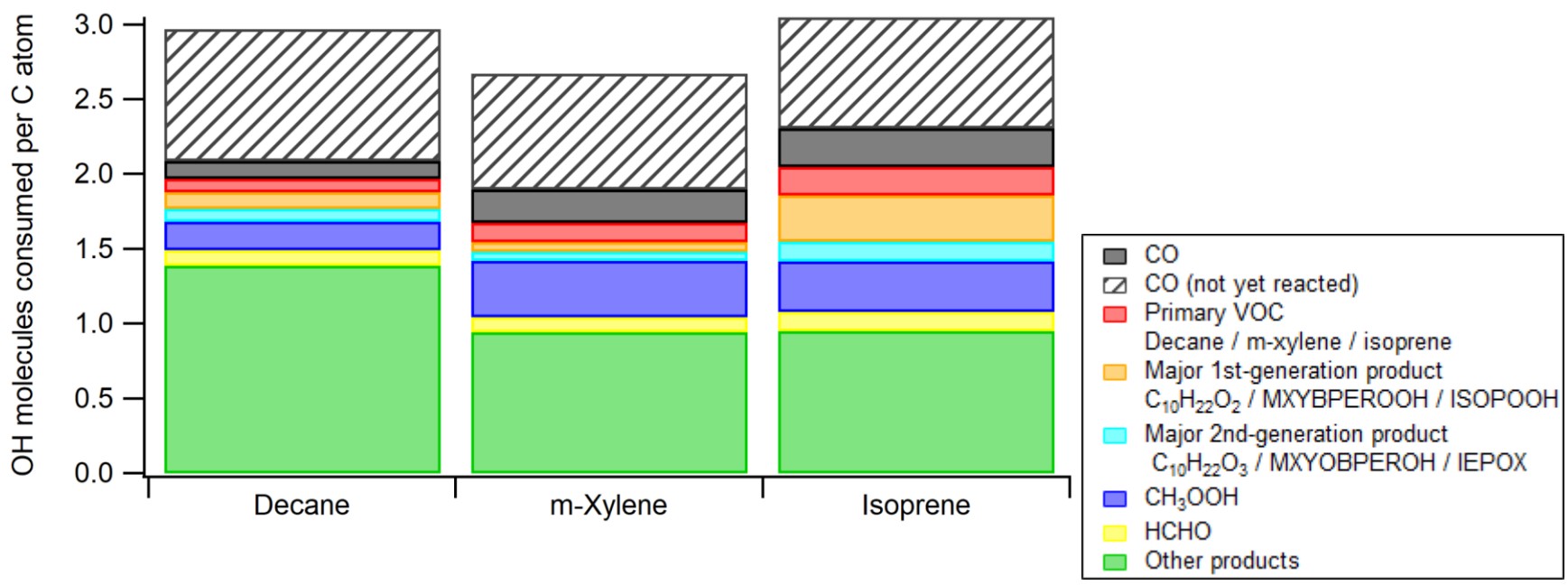

**Figure 6.** Average numbers of OH molecules consumed per C atom in the ambient cases with constant sunlight during photooxidation of isoprene, decane, and m-xylene. The contribution from CO that is not yet oxidized by OH at the end of simulation is also added to ensure that each CO molecule consumed one OH radical. ISOPOOH, IEPOX, $C_{10}H_{22}O_2$, and $C_{10}H_{22}O_3$ are isoprene hydroxyl hydroperoxides, isoprene epoxydiols, decyl hydroperoxides, and hydroxydecyl hydroperoxides, respectively. See Scheme S1 for the structures of MXYBPEROOH and MXYOBPEROH.

**Table 1.** Conditions and integration timesteps of the simulations in the present work.

| Precursor | Environment | Relative humidity (%) | UV | Initial OH reactivity ($s^{-1}$) | Integration timestep (s) |
|---|---|---|---|---|---|
| Methane | Ambient | 30 | Constant sunlight[a] | 10 | KinSim-determined[b] |
| | Oxidation flow reactor | 70 | High lamp setting[c] | | |
| Decane | Ambient | 30 | Constant sunlight[a] | 10 | Min: 0.1; Max: 120 (1 d), 300 (2–10 d) |
| | | | Diurnal sunlight | | |
| | Oxidation flow reactor | 70 | High lamp setting[c] | 10 | 0.0025 |
| | | 70 | Low lamp setting[e] | 10 | |
| | | 30 | Medium lamp setting[f] | 10 | |
| | | 3 | High lamp setting[c] | 10 | |
| | | 70 | High lamp setting[c] | 1000 | |
| | Chamber (gas-phase only) | 30 | CU Chamber spectrum[g] | 10 | Min: 0.1; Max: 120 (6 d), 300 (7–30 d if needed) |
| | | | | 100 | |
| | Chamber (gas-particle partitioning) | | | 10 | |
| | | | | 100 | |
| | Chamber (gas-particle-wall partitioning) | | | 10 | |
| | | | | 100 | |
| m-Xylene | Ambient | 30 | Constant sunlight[a] | 10 | Min: 0.1; Max: 120 (1 d), 300 (2–10 d) |
| | | | Diurnal sunlight[c] | | |
| | Oxidation flow reactor | 70 | High lamp setting[c] | 10 | 0.0025 |
| | | 70 | Low lamp setting[e] | 10 | |
| | | 30 | Medium lamp setting[f] | 10 | |
| | | 3 | High lamp setting[c] | 10 | |
| | | 70 | High lamp setting[c] | 1000 | |
| | Chamber (gas-phase only) | 30 | CU Chamber spectrum[g] | 10 | Min: 0.1; Max: 120 (6 d), 300 (7–30 d if needed) |
| | | | | 100 | |
| | Chamber (gas-particle partitioning) | | | 10 | |
| | | | | 100 | |
| | Chamber (gas-particle-wall partitioning) | | | 10 | |
| | | | | 100 | |
| Isoprene | Ambient | 30 | Constant sunlight[a] | 10 | Min: 0.1; Max: 10 (1 d), 120 (2–10 d) |
| | | | Diurnal sunlight[c] | | |
| | Oxidation flow reactor | 70 | High lamp setting[c] | 10 | 0.001 |
| | | 70 | Low lamp setting[e] | 10 | |
| | | 30 | Medium lamp setting[f] | 10 | |
| | | 3 | High lamp setting[c] | 10 | |
| | | 70 | High lamp setting[c] | 1000 | |
| | | 30 | Medium lamp setting[f] + 5x CU Chamber spectrum[g] | 10 | |

| | | | | | |
|---|---|---|---|---|---|
| | | 30 | Medium lamp setting[f] + 10000x CU Chamber spectrum[g] | 10 | |
| | Chamber (gas-phase only) | | | 10 | |
| | | | | 100 | |
| | Chamber (gas-particle partitioning) | 30 | CU Chamber spectrum[g] | 10 | Min: 0.1; Max: 10 (6 d), 120 (7–30 d if needed) |
| | | | | 100 | |
| | Chamber (gas-particle -wall partitioning) | | | 10 | |
| | | | | 100 | |
| Butane | Ambient | 30 | Constant sunlight[a] | 10 | Min: 0.1; Max: 120 (1 d), 300 (2–10 d) |
| Heptane | Ambient | 30 | Constant sunlight[a] | 10 | Min: 0.1; Max: 120 (1 d), 300 (2–10 d) |

[a] At solar zenith angle of 45°.

[b] Simulation performed in the solver KinSim, which fully controls its integration timestep selection.

[c] Diurnal variation between solar zenith angles of 0 and 90°.

[d] UV at 185 nm = $1 \times 10^{14}$ photons cm$^{-2}$ s$^{-1}$; UV at 254 nm = $8.5 \times 10^{15}$ photons cm$^{-2}$ s$^{-1}$.

[e] UV at 185 nm = $1 \times 10^{11}$ photons cm$^{-2}$ s$^{-1}$; UV at 254 nm = $4.2 \times 10^{13}$ photons cm$^{-2}$ s$^{-1}$.

[f] UV at 185 nm = $1 \times 10^{13}$ photons cm$^{-2}$ s$^{-1}$; UV at 254 nm = $1.4 \times 10^{15}$ photons cm$^{-2}$ s$^{-1}$.

[g] UV source spectrum of the University of Colorado Environmental Chamber Facility (Krechmer et al., 2017).