# Peer review of "Evolution of OH reactivity in low-NO volatile organic compound photooxidation"

_Atmospheric Chemistry and Physics, 2021_

## Author Response (AR1)

**Response to Reviews for the ACPD paper "Evolution of OH reactivity in low-NO volatile organic compound photooxidation investigated by the fully explicit GECKO-A model"**

We thank the referees for their reviews. To facilitate the review process, we have copied the reviewer comments in black text. Our responses are in regular blue font. We have responded to all the referee comments and made alterations to our paper (**in bold text**). Figures, tables, and sections in the responses are numbered as in the *revised* manuscript unless otherwise specified, while page and line numbers refer to the ACPD paper.

**Anonymous Referee #1**

R1.1) This study focuses on the investigation of the OH reactivity for several alkanes, m-xylene and isoprene by using fully explicit mechanisms generated with GECKO-A.

I find it rather difficult to appreciate the main goal and purpose of this study. The title is misleading and does not cover what is actually done within the study. The study is performed at zero NO, which is rather different than the mentioned "low-NO", where I think the majority of readers would assume this to mean below 100 pptv but not truly 0. The complete removal of NO from the model does not allow for nitrates, NO2 and ozone to be formed through the normal channels accessible at any atmospheric NO level. While there is nothing wrong in studying this type of environment, it should be clearly stated in the title/abstract/introduction/conclusion that this does not represent any real ambient condition. In a similar way, although not clearly stated, the impression I formed when reading the title was that the OH reactivity for different species would be investigated at ambient like conditions. This was again not found in the text, not only because of the lack of NO but also as most of the organic compounds chosen are very unlikely to be found in low-NO environment as they are mainly anthropogenic emissions. A first question raised is then why the focus is on zero NO instead of investigating more realistic ambient conditions?

To our knowledge, this is the first study of its kind, using the fully-explicit GECKO-A model to investigate the evolution of OH reactivity (OHR) for different compounds and reaction conditions. Therefore we chose to simplify the conditions of the study to enable easier comparison between the different systems. An alkane and an aromatic are included in this study as precursors, although they are usually anthropogenic, because the low-NO component of the chemistry in urban atmospheres has been increasingly important (Praske et al. 2018). In addition, intermediates of the oxidation of these precursors can still undergo processing under low-NO conditions in the atmosphere when the urban plume moves away from NO sources. We agree that carrying out similar studies for ambient conditions, especially in conjunction with experimental measurements, would be of high interest for future studies, but it was outside the

scope of this initial study. We have modified the conclusion to include this need for future studies (see the response to comment R1.2).

The title does not promise ambient-like conditions, and the abstract clearly explains what was done, so we have not changed the title in response to this comment.

Regarding low-NO conditions, those are generally understood to be the ones in which the $RO_2$ + NO reaction is of minor importance. Nitrates and ozone formation are also of minor importance in those cases. The difference between minor and zero contribution does not seem to be that important to us. Thus we believe that our results are indeed relevant to low-NO conditions.

R1.2) After reading the paper it became clear that the chosen reaction conditions are, in my opinion, more representative of what one can expect in oxidation flow reactors and certain simulation chambers, and the idea is to compare what happens in these laboratory devices against what would happen for similar conditions in ambient air. I think this a useful comparison, but I would recommend making that clearer in the title and introduction. I am stressing this point as in both the abstract and in the conclusion there is a strong remark about how OH reactivity is poorly constrained in the atmosphere, which is true, and how this study address, to some extent, all three issues raised by Williams and Brune (2015). I would recommend to state which these three issues are as I feel two main issues raised in the roadmap by Williams and Brune (2015) were the broader availability of instrumentation to measure OH reactivity, as well as improving and comparing the techniques measuring OH reactivity, neither of which was addressed in the current study. I assume this study refers mostly to the issue of the so-called "missing" OH reactivity often observed in field studies when comparing the measured OH reactivity with what can be calculated from the measured trace gases. If this is the case, I would recommend being more clear in stating it was this that is being addressed. This study does not include measured OH reactivity but only simulated OH reactivity and it investigates the OH oxidation of single compounds at atmospherically not relevant conditions. I feel the statement "largely speciated the likely source of the missing reactivity" seems a bit of an overstatement. Despite the use of a full explicit mechanisms, in the plots provided there are often lumped multifunctional species which do not clarify about which species contribute to what amount to the OH reactivity. Wouldn't it be possible to actually list the individual species and how much they contributes, or a list of the most prominent ones making up the largest fraction of the total? After all, many studies in the field showed that when a modelling of the OH reactivity is performed, the presence of large quantities of unmeasured oxygenated VOC (precursors to or actual multifunctional species) help in strongly reducing if not closing the gap between measured and calculated OH reactivity (Yang et al., 2016).

We thank the Referee for the input. First we quote the original text of the Williams and Brune (2015) perspective paper raising the 3 issues about OHR studies:

"*On 13-15th October 2014 a meeting was held at the Max Planck Institute for Chemistry in Mainz aimed at uniting for the first time the community of scientists interested in OH reactivity. Three key issues emerged. Firstly, significant missing reactivity is being found near ground level*

*and in particular in forested environments. Secondly, models are unable to satisfactorily simulate measured total OH reactivity and it is not clear whether this is due to incomplete chemical mechanisms (including unmeasured species and reaction rates) or the relatively unconstrained gas-to-surface deposition terms. Thirdly, even in clean remote regions over the Pacific Ocean missing reactivity can be appreciable.*"

In short, they are about i) missing reactivity, ii) model limitations, and iii) remote regions. None of these issues explicitly concern instrumentation. We believe that we indeed have addressed, to some extent, these issues in the Williams and Brune paper. Regarding missing reactivity, we suggest that much of it may be due to multifunctional species that were often not measured in past studies. To address model limitations, we have used a fully explicit model and discussed wall partitioning in chambers, which is one of "*the relatively unconstrained gas-to-surface deposition terms*" in chambers. In the last paragraph of the paper, we have discussed how the common trends of the evolution of OHR per C atom observed may help better constrain OHR in remote regions.

To clarify how what we have done in this study relates to the issues raised by Williams and Brune (2015), we have rephrased the last paragraph of the conclusions to read:

"**With all the key findings in this study presented above, we believe that we have, to some extent, addressed the issues of "missing reactivity", of model limitations, and of OHR in remote areas for OHR studies raised by Williams and Brune (2015). With the fully explicit GECKO-A model, we speciated the likely source of the "missing reactivity", i.e., multifunctional OVOCs. A contrast between the technical issues in some isoprene and m-xylene simulations and the high consistency in the other cases highlights the importance of the completeness of the mechanism (even beyond the MCM level) in OHR-related modeling studies. Substantial wall partitioning of OVOCs in some chamber experiments highlights the importance of better constraining "gas-to-surface deposition terms". More studies, both modeling (with highly chemically explicit mechanisms) and experimental (particularly low-NO), are needed to achieve better model-experiment closure. Finally, this study may have opened up the possibility of parameterizing the OHR evolution in (at least low-NO) VOC photooxidation as a function of $OH_{exp}$ only with the often-available knowledge on the first- and second-generation products and the relevant SARs such as Kwok and Atkinson (1995), as the OHR evolution beyond multifunctional species has been shown to be similar for most VOC oxidations. This parametrization may be utilized in regional and global models to better constrain OHR at high equivalent photochemical ages, e.g., in remote regions.**"

To respond to the Referee's request and further support that we have speciated OHR with fully explicit GECKO-A, we have added a table in the Supplement that reports the major OHR contributors in different cases. We reproduce the table below:

"**Table S2. Major contributors (>0.1 s$^{-1}$ or the first 20) to the OH reactivity at 1x10$^{10}$, 1x10$^{11}$, and 1x10$^{12}$ molecules cm$^{-3}$ s in the ambient case with constant UV of the photooxidation**

of decane, m-xylene, and isoprene. Also shown is the percentage of OH reactivity accounted for by the species reported in this table. Positional isomers with identical rate constant with OH are lumped into a single species. See Scheme S1 for the structures of the special names for m-xylene oxidation in this table. The "C1-O-C1" string in the names represents an epoxide structure. The isoprene case has no species contributing $>0.1$ $s^{-1}$ to the total OH reactivity at $1 \times 10^{12}$ molecules $cm^{-3}$ s, and is thus not shown for this OH exposure.

| Precursor | OH exposure (molecules $cm^{-3}$ s) | Species name | OH reactivity ($s^{-1}$) | Percentage of OH reactivity accounted for |
|---|---|---|---|---|
| decane | $1 \times 10^{10}$ | decane | 8.97 | >99 |
| | | decyl hydroperoxide | 1.95 | |
| | | dihydroperoxy decane | 0.12 | |
| | $1 \times 10^{11}$ | decyl hydroperoxide | 5.06 | 94 |
| | | decane | 3.35 | |
| | | dihydroperoxy decane | 2.17 | |
| | | hydroperoxy decanone | 1.65 | |
| | | decanone | 1.18 | |
| | | dioxodecyl hydroperoxide | 0.36 | |
| | | hydroxydecyl hydroperoxide | 0.27 | |
| | | dioxodecane | 0.14 | |
| | | hydroxydecyl dihydroperoxide | 0.13 | |
| | | hydroperoxyhydroxydecanone | 0.11 | |
| | $1 \times 10^{12}$ | $CH_3OOH$ | 0.88 | 53 |
| | | HCHO | 0.63 | |
| | | $CH_3CH_2OOH$ | 0.36 | |
| | | $CH_3C(O)OOH$ | 0.31 | |
| | | $CH_3CHO$ | 0.31 | |

| | | | | |
|---|---|---|---|---|
| | | HOOCH$_2$C(O)OOH | 0.23 | |
| | | CH$_3$C(O)CH$_2$CH$_2$OOH | 0.23 | |
| | | CH$_3$C(O)CH$_2$OOH | 0.21 | |
| | | HOOCH$_2$CH$_2$OOH | 0.20 | |
| | | HOOCH$_2$COOH | 0.20 | |
| | | CO | 0.19 | |
| | | HOOCH$_2$CHO | 0.17 | |
| | | CH$_3$C(O)CH$_2$CHO | 0.17 | |
| | | HOOCH$_2$CH$_2$C(O)OOH | 0.13 | |
| | | HOOCH$_2$CH$_2$COOH | 0.12 | |
| | | CH$_3$CH$_2$C(O)CH$_2$CH$_2$OOH | 0.11 | |
| m-xylene | 1x10$^{10}$ | m-xylene | 7.92 | 96 |
| | | MXYBPEROOH | 2.99 | |
| | | MXYEPOXMUC | 1.41 | |
| | | MXYOBPEROH | 1.13 | |
| | | m-xylenol | 0.90 | |
| | | CH$_3$C(O)CH(OOH)CH(OH)C1H-O-C1(CH$_3$)CHO | 0.34 | |
| | | dimethyl catechol | 0.22 | |
| | | MXYOLOOH | 0.17 | |
| | 1x10$^{11}$ | methylglyoxal | 1.04 | 69 |
| | | m-xylene | 0.96 | |
| | | CH$_3$OOH | 0.91 | |
| | | MXYOBPEROH | 0.81 | |
| | | MXYBPEROOH | 0.59 | |
| | | CH$_3$C(OOH)(CHO)$_2$ | 0.45 | |

| | | | | |
|---|---|---|---|---|
| | | CH$_3$C1(CHO)-O-C1HCHO | 0.43 | |
| | | CH$_3$C(O)CH=CHC1H-O-C1(CH$_3$)CHO | 0.39 | |
| | | HCHO | 0.38 | |
| | | CH$_3$C(O)CH(OOH)CH(OH)C1H-O-C1(CH$_3$)CHO | 0.34 | |
| | | CH$_3$C(O)OOH | 0.34 | |
| | | HOC(O)C1H-O-C1(CH$_3$)CHO | 0.31 | |
| | | glyoxal | 0.22 | |
| | | m-xylenol | 0.22 | |
| | | HOOC(O)CH=C(CH$_3$)CHO | 0.21 | |
| | | CH$_3$C(O)CH=CHC(O)OOH | 0.20 | |
| | | CH$_3$C(O)C(O)CH(OH)C1H-O-C1(CH$_3$)CHO | 0.18 | |
| | | TT801J | 0.17 | |
| | | TT8004 | 0.16 | |
| | | CH$_3$O$_2$ | 0.15 | |
| | 1x10$^{12}$ | CH$_3$OOH | 0.48 | 56 |
| | | CO | 0.19 | |
| | | HCHO | 0.12 | |
| isoprene | 1x10$^{10}$ | isoprene | 3.68 | 93 |
| | | ISOPOOH | 2.80 | |
| | | HOOCH$_2$CH=C(CH$_3$)CHO | 0.23 | |
| | 1x10$^{11}$ | ISOPOOH | 0.37 | 36 |
| | | IEPOX | 0.33 | |

"

We do not further modify the text together with the addition of this table, but cite it wherever appropriate in the text.

R1.3) Finally, I think there should be a better description of which simulation chambers (type, size, …) this study refers to. The same holds for the oxidation flow reactors (OFR) discussed. Specifically: atmospheric simulation chambers can be as small as few m3 up to hundreds of m3 and their size and residence time of gases in the chamber and initial concentrations of reagent will strongly influence how lower-volatility compounds are lost on walls. Within this study only a value without wall effect, and one with a certain loss on walls of certain compounds is given but no explanation as to which chambers this could apply to. The largest atmospheric simulation chambers such as SAPHIR (Rohrer et al., 2005), Jülich, Germany, or EUPHORE (Siese et al., 2001), Valencia, Spain are much less affected by wall losses as compared to smaller chambers. In addition, this study seems to focus on simulation chambers where high concentrations of reagents and intermediates are used. This is again not what is investigated in chambers such as SAPHIR and EUPHORE where experiments are conducted with atmospheric like concentration of reagents and intermediates, and where indeed no large discrepancies have been observed between measured and calculated/modelled OH reactivity; see e.g. (Fuchs et al., 2013; Nölscher et al., 2014; Fuchs et al., 2017; Novelli et al., 2018). One of the reasons for the good agreement between measured and calculated/modelled OH reactivity found for these studies is clearly the lack of large amounts of oxygenated products, so these studies might not be representative examples for the topic of this study where the oxidation in the chamber is simulated up to 10 days. A more clear specification up front as to what type of experiments, which type of chambers, which reaction times, and which bracket of reaction conditions is examined would avoid the incorrect interpretation that this study applies to all chambers and most chamber studies.

We have added the characteristics of the chamber being simulated (the CU Chamber) to the paper (see the response to comment R2.5). The CU Chamber is of medium size in the typical range of chambers in use (~20 m$^3$). EUPHORE and SAPHIR have a smaller surface-to-volume ratio only by a factor of ~2. Also, the default initial OHR (10 s$^{-1}$) in this study is comparable to that in the SAPHIR experiments in Nehr et al. (2014), of which some OH and OHR data have been made available to us. We thus believe that the conclusions about wall partitioning in the CU Chamber should approximately apply to some SAPHIR (and EUPHORE) experiments, too.

To further support this claim, we compare the OHR evolution in a few m-xylene model cases in this study with that measured during the p-xylene photooxidation experiments from Nehr et al. (2014). m-Xylene and p-xylene are very similar precursors, the initial OHR is equal (close) to 10 s$^{-1}$ in the model (experiments), and NO level did not show a strong impact on the OHR evolution measured (Fig. R1). The OHR evolution observed by Nehr et al. (2014) is very close to that simulated for the CU chamber (with wall) case in terms of both the OHR relative peak height and the photochemical age of its occurrence (a few hours, far shorter than 10 days), while the OHR peak observed by Nehr et al. (2014) is much lower than that in the modeled case for the ambient and CU Chamber (no wall) cases (Fig. R1). The large difference around the OH$_{exp}$ of the OHR peak cannot be adequately explained by the condensation of OVOCs condensed onto

aerosols which suggests that significant wall partitioning of OVOCs may have also occurred in SAPHIR.

[Figure]

*Figure R1. Gas-phase OH reactivity as a function of OH exposure in the cases of m-xylene photooxidation at zero NO in the CU chamber (without aerosol or wall partitioning; with aerosol partitioning; and with wall and aerosol partitioning), in the atmosphere (constant and diurnal UV) and in the experiments of p-xylene photooxidation in the high-NO (cyan) and high/low NO transition (green) regimes in the SAPHIR chamber for Nehr et al. (2014).*

The studies that the Referee mentioned as not having observed discrepancies between measured and modeled OHR (Fuchs et al., 2013; Nölscher et al., 2014; Fuchs et al., 2017; Novelli et al., 2018) all concern photooxidation of isoprene and related compounds (C5 or lower). We have shown that the wall partitioning of these species is much less significant than for the m-xylene- and decane-derived species (Figs. 3, S5, and S6) (see also the response to comment R2.2). Up to $OH_{exp}$ of a few $10^{10}$ molecules $cm^{-3}$ s (roughly the highest OH exposure achieved in these experiments), at which almost all isoprene is oxidized and when saturated multifunctional species (e.g. IEPOX) have been significantly produced (Fig. S6), the OHR of the products lost to the wall is smaller than that of the products remaining in the gas phase. Compared to the total OHR, the OHR lost to the wall is even more minor. In these studies mentioned by the Referee, total OHR measurement techniques, usually with relatively large uncertainties, may have been unable to distinguish this minor fraction of OHR lost to the wall, when compared to modeling results. Measurements of individual OHR contributors in these

studies focused on the well-known products such as ISOPOOH, IEPOX, MVK, and MACR, which are not highly functionalized enough to be of low volatility and partition substantially to the walls (Krechmer et al., 2015). It is expected that good agreement between gas-phase measurement and modeling can be achieved for these species. As we show above, for larger precursors that can easily produce lower-volatility products (e.g. xylene), wall partitioning of oxidation products can be substantial.

In addition, neither techniques that speciate gas-phase compounds (e.g. PTR-MS) nor those measuring total gas-phase OHR (e.g. LIF) are able to identify OHR lost to the wall. Agreement between measured total OHR in a chamber and OHR calculated from speciated individual species in that same chamber does not support lack of wall partitioning.

We have added the following text to L120 to address the effect of chamber size and the applicability of wall losses to modeling chambers:

**"Wall partitioning in chambers at equilibrium is a function of the surface-to-volume ratio (Krechmer et al., 2016). The timescale to approach equilibrium is expected to be larger in larger chambers, but still far shorter than the long experiments needed to investigate high photochemical ages. Therefore differences in wall partitioning timescale are not important for this study. Figure S9 of Krechmer et al. (2016) compared the CU Chamber and a few other well-known chambers (including very large ones such as EUPHORE (Siese et al., 2001) and SAPHIR (Rohrer et al., 2005)), showing relatively small differences (within a factor of ~2 in terms of surface-to-volume ratio). Therefore the conclusions about wall partitioning in this study should be approximately applicable to most chambers."**

R1.4) The paper is well written but rather than recommending publication I suggest the authors consider the points mentioned above and attempt to better focus the study in such a way that makes it more clear how ambient OH reactivity values and the issue of "missing' OH reactivity is addressed by their modeling. In its current form, this link is not clear and while the data is interesting there seems to be little direct applicability for it.

Please see the response to comment R1.2.

Specific comments:

R1.5) Abstract. The first sentence needs a bit of explanation of where it is poorly constrained. In every environment? In chamber experiments? For specific chemical regimes? What is the reason behind the chosen VOCs?

We have modified the first sentence of the abstract as follows:

"**OH reactivity (OHR) is an important control on the oxidative capacity in the atmosphere but remains poorly constrained in many environments, such as remote, rural, and urban atmospheres, as well as laboratory experiment setups under low-NO conditions.**"

We selected one representative species for each of the three main types of primary VOCs, i.e., alkanes, alkenes, and aromatics. We think that this message has already been implicitly conveyed in the following sentence in the abstract: "*We use the fully explicit Generator of Explicit Chemistry and Kinetics of Organics in the Atmosphere (GECKO-A) model to study the OHR evolution in the low-NO photooxidation of several VOCs, including decane (an alkane), m-xylene (an aromatic), and isoprene (an alkene).*" Therefore, we do not modify the abstract to give additional justification for the chosen VOCs, given the need for brevity.

R1.6) Page 2, Line 57. I do not understand the use of the Nehr et al. (2014) study here. It is true that the OH reactivity was measured but it is not compared with a calculated or measured OH reactivity. The studies highlighted above (Fuchs et al., 2013; Fuchs et al., 2017; Novelli et al., 2018) would be in my opinion a better choice.

We have changed the cited references here as suggested by the Referee.

R1.7) Page 6 Lines 223-230. It is not specified what the basis is for the implementation of the mechanisms. There is no reference to any previous study or to SAR. How is the rate coefficient between OH and MXYLOOH estimated to be ~ 3e-11 cm3 s-1? I would also refrain from using the notation of kOH for the rate coefficient as many groups use that same notation for the OH reactivity. The same question applies for the estimated reaction rate of RO2+HO2. Can a reaction with O3 compete with the reaction paths normally available for alkoxy radicals? What is the rate of the reaction with O3 and the alkoxy and could also reaction with OH compete? Are there appropriate references for adding this type of chemistry? In the schemes I would recommend keeping the orientation of the molecules always the same to enhance readability. E.g. MXCATEC10 has the methyl group pointing up, yet after reacting with O3 the methyl group points down. As I understood there is zero NO in the model, but then how is the O3 formed, as it is not stated explicitly that it is added as a reactant for all simulations? The OH in the OFR is produced from photolysis of O3 (which is a hint O3 is added as a reactant), which raises the question on how is OH produced in the ambient and the chamber cases? It would be good to describe these critical inorganic reactions, as well as how much O3 is present in the different simulations.

We have changed "$k_{OH}$" to "$k_{VOC+OH}$" in L226.

We have renamed MXYLOOH to MXYOLOOH throughout for consistency with the MCM.

We have adjusted the orientations of some molecular structures in Scheme S1. as suggested by the Referee. During this process we discovered that we had inadvertently included a previous version of Scheme S1. In the version of the m-xylene mechanism used in the final simulations for this paper, we have 1) allowed GECKO-A to use its SARs to replace the MCM reactions of MXYOLO2+OH and MXYOLOOH+OH with SAR-generated OH reactions that lead to (mainly) bicyclic diether unsaturated peroxy radicals, 2) added a new reaction MXYCATECH+OH analogous to the MCM reaction of xylenol+OH→MXYOLO2, and 3) allowed GECKO-A to use its SARs to insert the reaction MXY1OOH+OH which is not included in the MCM. To better describe the SARS involved, we have added a reference (Lee-Taylor et al., 2015), which details several updates to SARs in GECKO-A. The justification and references for each of these reactions is described in new text between L138-141 and L145-148 as follows:

"**GECKO-A (Aumont et al., 2005; with updates as described by Camredon et al., 2007; Valorso et al., 2011; Lee-Taylor et al., 2015), is an explicit chemical model which uses known mechanisms and rates supplemented with experimentally-based structure-activity relationships to generate comprehensive atmospheric oxidation mechanisms for organic species.**"

"**The core isoprene scheme in GECKO-A is adopted from the Master Chemical Mechanism v3.3.1 (Jenkin et al., 2015), while the meta-xylene oxidation mechanism follows MCM v3.2 (Jenkin et al, 2003, Bloss et al, 2005), typically until ring-breaking occurs, whereupon the GECKO-A mechanism generator implements the standard SAR protocols as described by Aumont et al. (2005), Camredon et al. (2007), and Lee-Taylor et al. (2015). Under the zero-NO conditions employed in this study, we find that, in two of the four m-xylene reaction channels (xylenol, 17%; and MXYLO2, 4%), some product species persist anomalously owing to lack of alternative reaction pathways in the MCM. We therefore allow GECKO-A to apply the standard SARs to two cyclic non-aromatic products of xylenol (MXYOLO2 and MXYOLOOH in the 51% xylenol OH-oxidation channel, see Scheme S1). We also introduce OH-oxidation of MXYCATECH and MXY1OOH (in the 42% and 7% xylenol OH-oxidation channels), and of MXYLOOH and MXYLAL (in the MXYLO2 channel), assuming similarity to the MCM OH-oxidation of xylenol to MXYOLO2, and with net OH rate constants estimated using the EPA EPISuite software package (US EPA, 2012). MXYLOOH, MXCATECH and MXYLAL each yield between two and six bicyclic non-aromatic substituted peroxy radicals, with net OH rate constants of $1.77 \times 10^{-11}$, $1.56 \times 10^{-10}$ and $8.6 \times 10^{-13}$ cm$^3$ molecule$^{-1}$ s$^{-1}$ respectively. (The MXYLOOH OH-rate also includes MXYLAL production). MXY1OOH is assigned a substituted single-ring hydroxy-ketone product, with OH rate constant $3.26 \times 10^{-11}$ cm$^3$ molecule$^{-1}$ s$^{-1}$. The early part of the meta-xylene reaction scheme used in this work is shown in Scheme S1.**"

The updated and corrected scheme is included in the SI and is reproduced below.

"**Scheme S1. Mechanism of xylenol photooxidation in the absence of NO, derived from MCM v3.2 (Jenkin et al., 2003; Bloss et al., 2005). Panel a): All meta-xylene reaction channels; Panel b): Xylenol reaction channels, expanded. Numbers refer to branching ratios. Green text indicates products from MCM v3.2; blue text indicates products generated by GECKO-A and added in this study; blue-and-green indicates products provided by both MCM3.2 and GECKO-A. Species whose products persist anomalously under zero-NO conditions are indicated with green dashed borders. Short dotted arrows indicate chemistry continuing according to standard SARs. Dashed arrows indicate photolysis pathways (which are usually minor). For illustration, we extend the dominant (HO$_2$ and higher branching ratio OH) oxidation pathway of one of the most prevalent RO$_2$ products (2T800J: panel b, lower center to upper right). Its main pathway from initial OH oxidation (lower right, 2nd row up) is also extended to show an example of typical ring-breaking chemistry. Net OH rate constants for MXYLOOH, MXY1OOH, MXYCATECH, and MXYLAL are 1.77x10$^{-11}$, 3.26x10$^{-11}$, 1.56x10$^{-10}$ and 8.6x10$^{-13}$ cm$^3$ molecule$^{-1}$ s$^{-1}$ respectively. (The MXYLOOH OH rate includes MXYLAL production). Figure adapted from MCM website, http://mcm.leeds.ac.uk/MCMv3.3.1, accessed July 3, 2020.**

[Figure]

Note that this scheme no longer relies upon $O_3$ reactions, which we agree with the reviewer are unlikely to be competitive in our simulations. To avoid further complications in the simulations, we did not add any initial $O_3$. All $O_3$ was formed during the model runs. In the ambient and chamber cases, $O_3$ production is dominated by the reactions of acylperoxy and $HO_2$, and there is typically a few ppb of $O_3$ in these cases. In the OFR cases, as UV at 185 nm is used, $O_2$ is photolyzed at this wavelength to produce $O(^3P)$, which then recombines with $O_2$ to form $O_3$. The $O_3$ concentrations in the OFR cases range from a few ppb to ~1 ppm. The reactions of $O_3$ are generally not important in this study. This is also true in the OFR cases, as their $HO_x$ levels are also higher than in the atmosphere. The species in Scheme S1 whose reactions with $O_3$ are

important for their loss are all in minor pathways, i.e., MXYCATECH and MXY1O (with a total branching ratio <0.09).

In accordance with the updated m-xylene mechanism, we have rerun all the m-xylene simulations to better deal with the issue of some species persistent at very high ages due to lack of oxidation pathways. We have also updated the text in the following places to reflect the new m-xylene results:

At L462:

"**However, in the OFR cases with strong water vapor photolysis (not in the other OFR cases), the third step does not play a significant role and $\beta_1$ is ~0 at very low $OH_{exp}$ (Fig. 1) due to the relatively slow second step ($RO_2 + HO_2$).**"

And at L470:

"**As more multifunctional species are formed (particularly through ring-opening) near the $OH_{exp}$ of the peak $OHR_{VOC}$, $HO_x$ recycling is also active, with $\beta_1$ increasing and $\beta_2$ remaining high (Fig. 1). There is a high peak in $\beta_2$ for the chamber case with high initial OHR (100 $s^{-1}$) and no aerosol or wall partitioning. It results from $RO_2$ cross-reactions, many of which produce alkoxy radicals that subsequently yield carbonyls and $HO_2$ through reactions with $O_2$ (Orlando and Tyndall, 2012). $RO_2$ cross-reactions are significant in that $OH_{exp}$ range also because i) high precursor concentration translates into higher $RO_2$ concentration and ii) acylperoxy radicals, whose reactions with other $RO_2$ are fast (Orlando and Tyndall, 2012), are rapidly formed from the oxidation of -CHO groups in the ring-opening products (Scheme S1). The chamber case with high initial OHR and gas-particle-wall partitioning does not have such a high $\beta_2$ peak, because of fast partitioning of the oxidation products containing -CHO groups to the aerosol and wall phases, which significantly reduces acylperoxy radical concentration around the $OH_{exp}$ of the peak $OHR_{VOC}$. At higher $OH_{exp}$, calculated $\beta_1$ and $\beta_2$ become less reliable, since remaining apparent OHR contributors may in fact be persistent artifacts of the incompleteness of the (hand-written) m-xylene oxidation mechanism which may substantially bias $\beta_1$ and $\beta_2$ when the concentrations of remaining OHR contributors should be generally low. Therefore, we do not try to interpret the features in $\beta_1$ and $\beta_2$ at high $OH_{exp}$ for m-xylene oxidation.**"

And at L608:

"**The number of OH consumed per C atom in m-xylene oxidation is slightly lower than 3 (Fig. 6) because of the multiple addition of $O_2$ following a single OH addition in the initiation reaction, i.e., m-xylene + OH.**

[Figure]

[Figure]

**Figure 6. Average numbers of OH molecules consumed per C atom in the ambient cases with constant sunlight during photooxidation of isoprene, decane, and m-xylene. The contribution from CO that is not yet oxidized by OH at the end of simulation is also added to ensure that each CO molecule consumed one OH radical. ISOPOOH, IEPOX, $C_{10}H_{22}O_2$, and $C_{10}H_{22}O_3$ are isoprene hydroxyl hydroperoxides, isoprene epoxydiols, decyl hydroperoxides, and hydroxydecyl hydroperoxides, respectively. See Scheme S1 for the structures of MXYBPEROOH and MXYOBPEROH.”**

R1.8) Section 2.3. Such a tool would be extremely useful for the atmospheric community. Do the authors plan to share it?

We thank the Referee's call to publicly release GECKO Loader and Plotter in the interest of the atmospheric chemistry community. We do intend to release it as a community tool, as we have

done with other pieces of software. However, that cannot be done yet due to the need for additional development and cleanup, as well as limited manpower to support the tool.

R1.9) Page 7, Line 270. For which temperature and pressure is the rate coefficient between OH and CO valid? Also, at which temperature were the simulations performed?

We always set the temperature to 295 K and the atmospheric pressure to 835 mbar in the simulations. We have added the following sentence to L127 to include this information:

"**As several key parameters of the chamber and OFR cases were obtained experimentally at room temperature and atmospheric pressure in Boulder, Colorado, USA (typically 295 K and 835 mbar), for an easier comparison, we use these values for the temperature and atmospheric pressure of all model cases.**"

The rate coefficient of CO+OH is also for this temperature and pressure. Note that CO+OH has little temperature dependence between 200-300 K and its rate coefficient decreases only by ~30% at very low pressures (IUPAC Task Group on Atmospheric Chemical Kinetic Data Evaluation). We believe that the value of the rate coefficient of CO+OH used in this study is robust and do not modify the text to discuss it further.

R1.10) Page 8, line 310. I do not understand what the notation "<x3 more rapidly" stands for.

The text that the Referee quoted means less-than-3-times more rapidly in terms of the rate coefficient of the whole molecule. We have modified this sentence for clarity as:

"**The main first-generation products, i.e., secondary decyl hydroperoxides, react with OH only less-than-3-times more rapidly (in terms of the rate constant of the whole molecule) than does decane, as the significant activation effect of the –OOH group only applies to the α-H, and all other H atoms in this long chain alkyl, though less reactive, can be abstracted by OH (Kwok and Atkinson, 1995; Aumont et al., 2005).**"

R1.11) Page8, line 312. I think it should be "can be abstracted"?

We have modified the text as suggested by the Referee. Please see the modified text in the response to comment R1.10.

R1.12) Page 9 Lines 323 and 324. I do not feel this sentence is necessary; I would go ahead and explain what the reasons are without saying it is different reasons…

We have removed this sentence as suggested by the Referee.

R1.13) Page 10, Lines 379 and 380. I think realistic and unrealistic are not very good in constraining how different simulations chambers are affected by partitioning of volatile compounds on the walls. As mentioned above explanations about which chambers/type of chambers this study is referring to and how these are positioned within the variety of chambers used is needed to avoid giving the wrong impressions that all simulation chambers have (huge?) wall losses.

Please see the response to comment R1.3. We believe that our characterization is correct, within the current state of the science, for most well-known chambers in this field. We have modified this sentence as follows:

"**Unfortunately, the lack of wall partitioning is not realistic for current chambers.**"

R1.14) Summary and conclusions. I feel that bullet point 3 and 4 on page 16 don't belong to the conclusion. Aside from the impression they are saying the same, they are also merely the result of the modelled used. GECKO-A uses SAR and therefore explicitly incorporated that C atoms in >C=C<, -CH2-, and -CH3 have OHR per C atom on the order of 10-11, 10-12, and 10-13 cm3 s-1 ; this "conclusion" is thus a given and does not seem to add much to the discussion, nor it is a new finding as it is well established in the SARs. As there is a long discussion about how wall losses can impact the findings of chamber experiments and need to be carefully accounted, I recommended once more to be clear about which chambers/type of chambers the study applies to, to avoid misinterpretation. As already mentioned above, I disagree that this study addresses the issues raised by the roadmap on OH reactivity by Williams and Brune (2015). As the paper in its current form has little applicability to real atmospheric conditions it is difficult to claim it helps understanding the source of the "missing" OH reactivity observed in various field studies.

We prefer to keep the 3rd and 4th bullet points. They may not be conclusions, but are indeed a summary of a few key results of this study. The 3rd point discussed how the OHR evolves in the cases of different precursors and why OHR increases then decreases for some precursors while it always decreases for others. The 4th point discusses why OHR per C atom for different precursors converges and evolves in a similar way afterwards, which may have implications for global OHR modeling.

Nevertheless, we agree with the Referee that "*C atoms in >C=C<, -CH$_2$-, and -CH$_3$ have OHR per C atom on the order of 10$^{-11}$, 10$^{-12}$, and 10$^{-13}$ cm$^3$ atom$^{-1}$ s$^{-1}$*" is well established in the SARs and thus remove it. The first sentence of the 4$^{th}$ bullet point now reads as:

"**A relatively weak enhancement of OHR per C atom of a C atom with -OOH substitution can explain the large range spanned by the precursors and their intermediates/products in this study at low OH$_{exp}$.**"

Regarding the discussion about wall partitioning, we believe that it is generally appropriate (please see the response to comment R1.3 for further details). For added clarity, we have modified the text to L659 to read:

"**In current chambers, gas-wall partitioning can be a prominent issue that causes substantial wall partitioning of certain OVOCs of lower volatility, depending on the chemical system under study. The clearest example in this study is the substantial wall losses of C10 multifunctional species from the gas phase in decane oxidation, and hence the remarkably lowered OHR$_{VOC}$ peak height in the chamber simulation.**"

And to L669 to read:

"**Systematic OVOC gas-particle-wall partitioning corrections must be made for low-NO oxidation chamber experiments that study OHR$_{VOC}$. In case of large precursors, highly chemically explicit modeling will likely be necessary to infer the OHR of multifunctional species, which may account for a large fraction of missing reactivity but suffer substantial wall losses. Although the few existing chamber studies on OHR$_{VOC}$ evolution were all under high-NO conditions, which may result in more fragmentation and higher-volatility products, the magnitude of wall partitioning of large multifunctional species in this study is so substantial that we believe this effect would also be important at high NO. Schwantes et al. (2017) considered wall partitioning in their modeling of o-cresol oxidation based on MCM v3.3.1 but still could not achieve good agreement with the measurements for a number of products. Considering this, one should not assume that it is appropriate to neglect gas-particle-wall partitioning in high-NO chamber experiments, just based on agreement between the high-NO chamber experiments and modeling with MCM-based schemes and without gas-particle-wall partitioning corrections. Even for OHR studies with less surface loss issues, e.g., ambient studies, a combination of gas-phase-only OHR measurement and modeling may still not be adequate as reduction of OHR due to OVOC condensation on aerosols can also be significant in some situations (Fig. S4). Therefore, condensed phases (particle and wall) need to be included in future OHR studies to better assess the deviation of the actual OHR from a purely gas-phase picture.**"

For the text related to Williams and Brune (2015) in the Summary and Conclusions section, please see the response to comment R1.2.

**Anonymous Referee #2**

R2.1) In this manuscript, the authors analyzed OH reactivity during oxidation of three types of volatile organic compounds by using a fully explicit chemical mechanism, GECKO-A. The authors focused on the simulations under the low NO condition.

One of my concern is that simulation results are reported in this manuscript without any validation. Of course, I understand that experimental studies of OHR under low NO condition have not been conducted, but this point should be further examined in future studies.

We have added some text in the conclusions that calls for low-NO chamber experiments for OHR studies. Please see the response to comment R1.2 for the modified text.

R2.2) Another concern is that aerosol formation is not considered in this simulation. In Line 164: the authors showed that major intermediates/products under low NO condition have high C*. However, multi-functional compounds have important contributions to OHR as in Figures 3, S4, and S5, and vapors deposit onto chamber walls. Thus, I don't think gas-particle partitioning is negligible. The authors noted that aerosol concentration is very low under low-NO condition in the ambient air (remote atmosphere). However, not in the context of atmospheric analysis, but in the context of the interpretation of chamber experiments, contributions of gas-particle partitioning could be significant. Thus, in my opinion, the authors' demonstration with gas-wall partitioning but without gas-particle partitioning is misleading; contributions of gas-wall partitioning could be overestimated. I recommend the authors to consistently simulate gas-particle-wall partitioning, particularly for the chamber cases.

Even with these concerns, this manuscript includes useful information about the OHR of VOCs, as well as the oxidation pathways of VOCs. Thus, I recommend this manuscript for publication after my concerns adequately addressed.

We thank the reviewer for this comment. We have rerun the simulations for the wall-partitioning chamber cases with aerosol partitioning included, and we have also run the additional simulations for the chamber cases with aerosol partitioning only (no wall partitioning) to quantify the effects of different types of partitioning. With the current case settings, it is very difficult to constrain organic aerosol (OA) concentration to atmospherically relevant values since a time-dependent fast-changing dilution is necessary. Considering this difficulty and the usually low OA concentration in typical low-NO environments, we thus do not perform additional simulations for ambient cases with aerosol partitioning.

In the additional simulations, the initial aerosol seed concentration is ~2 µg m$^{-3}$. The particle phase still plays a minor role in these new runs around the OH$_{exp}$ of OHR peak and at lower OH$_{exp}$. At higher OH$_{exp}$, as sufficiently low-volatility species condense onto the aerosol, OA concentration increases and OA finally may compete with the walls as a sink of OVOC in the cases of larger precursors (i.e., decane and m-xylene). In isoprene oxidation, since not much OA can be formed, wall partitioning still dominates OVOC condensation.

In the process of conducting the aerosol simulations, we have discovered an error in the choice of parametrization of wall partitioning in the simulations of the chamber cases with walls reported in the ACPD paper. The original parameterization of Matsunaga and Ziemann (2010) had been used in the ACPD simulations, instead of the more recent and more accurate Krechmer et al. (2016) parameterization. We have rerun all the chamber simulations using the updated wall loss parameterization.

We have updated Figs. 2, 3, 4, S5, and S6. Panels for the aerosol phase in the chamber cases with gas-particle-wall partitioning have been added into Fig. 4. We have also added Fig. S4 to include the results of the chamber cases with gas-particle partitioning only (no wall). The updated and added figures are reproduced below.

"

[Figure]

**Figure 2. Ratios of OHR of the products present in the gas phase between the chamber cases without gas-particle-wall partitioning and i) with gas-particle (G vs. G+A) or ii) gas-particle-wall partitioning (G vs. G+A+W) at initial OHR of 10 s$^{-1}$, and between the ambient cases with constant and diurnal sunlight for the photooxidations of decane, m-xylene, and isoprene as a function of OH exposure.**

[Figure]

**Figure 3. Absolute and fractional contributions to the organic OHR during decane photooxidation of the main species and types of species as a function of OH exposure in the ambient case with constant sunlight; the OFR case with relative humidity of 30%, medium UV lamp setting, and initial OHR of 10 s$^{-1}$; and the chamber case with initial OHR of 10 s$^{-1}$ and gas-wall partitioning. The types of species shown in this figure exclude the C1 and C2 species listed separately. The OHR of**

the particle- and wall-phase species are the values as if those species are gas-phase OHR contributors, although they actually do not react with OH in the simulations."

"

[Figure]

**Figure 4. Average number of functional group per C atom as a function of OH exposure in the saturated multifunctional species in the ambient case with constant sunlight, the OFR case with relative humidity of 70%, high UV lamp setting, and initial OHR of 10 s⁻¹, and the gas, aerosol, and wall phases in the chamber case with initial OHR of 10 s⁻¹ and**

[Figure]

**gas-particle-wall partitioning of the photooxidations of decane, m-xylene, and isoprene. Note that the functional group "-O-" represents ether, ester, and epoxy groups in the GECKO-A model."**

**Figure S4. Same format as Fig. 1, but for the chamber cases without aerosol or wall, with aerosol (no wall), and with aerosol and wall for decane, m-xylene, and isoprene photooxidation."**

"

[Figure]

**Figure S5. Same format as Fig. 4, but for m-xylene photooxidation."**

"

[Figure]

**Figure S6. Same format as Fig. 4, but for isoprene photooxidation."**

We have also updated the relevant text at L34 in the abstract:

[revised manuscript text omitted]

We have also updated all other figures that need to include the results of the new simulations for the chamber cases with gas-particle-wall partitioning. As the changes are very minor, we do not show the rest of them in this response to reduce clutter.

R2.3) Line 72: Why is it difficult to achieve low-NO conditions in chambers?

Residual NO in chamber air, and/or HONO formation from chamber walls is very difficult to remove completely. Even a trace amount of NO at a few tens of ppt level can compete with $HO_2$ for $RO_2$ losses. Most commercial trace-level analyzers are not sensitive enough to measure these levels. For example, commercial chemiluminescence (CL) instruments for NO measurement have detection limits around 500 ppt. State-of-the-art CL instruments that can measure NO at tens-of-ppt level are needed (Nguyen et al., 2014). Therefore it is technically difficult to experimentally ensure that low-NO conditions are achieved in a chamber once it has been used for high-NO experiments.

To clarify this, we modify the text to L69 to read:

"**To our knowledge, no experiment of this type at low NO, where $RO_2$ can substantially react with hydroperoxy radical ($HO_2$), has been published so far, potentially partially due to the difficulty in experimentally ensuring that low-NO conditions are achieved in chambers (Nguyen et al., 2014).**"

R2.4) Line 130: Another than what? Explanation of a kinetic solver is not given so far.

We have modified the relevant sentence at L129 to clarify this. The updated sentence reads:

"**Note that these two simulations are performed using the GECKO-A generated mechanism (see Section 2.2) in KinSim (Peng and Jimenez, 2019), a chemical-kinetics solver that is not GECKO-A's default, to avoid possible numerical issues in the GECKO-A internal solver, as methane oxidation by OH is very slow (Atkinson and Arey, 2003) and very long runs are needed.**"

R2.5) Line 118: Parameters for the chamber should be explicitly given (chamber volume, area, and coefficient of eddy diffusion).

We have added the following sentence in L118 to include the information requested by the Referee (note that coefficient of eddy diffusion can be obtained from wall condensation time scale, which is more directly measurable):

"**The CU Chamber has a volume of ~20 m$^3$, a surface area of ~65 m$^2$, and an estimated wall condensation timescales of ~1000 s (Krechmer et al., 2016).**"

R2.6) Line 305: This is not the case at higher OH exposure (> 10^12 molecules cm-3 s) or for some of the sensitivity cases.

The sentence "*Contrary to the methane cases, OHR$_{VOC}$ in all five decane simulations for OFR conditions is lower than that for ambient conditions.*" at L305 was incorrect. We have removed it and thank the Referee for pointing it out.

R2.7) Figure 1: The authors conducted several sensitivity simulations for RH, UV, initial OHR, but they did not discuss the variability of these simulation results. In the course of my comment for Line 305, the argue of the authors is not necessarily true for all the sensitivity cases. If the authors leave results of these sensitivity simulations, rationale and some discussion of these sensitivity simulations are necessary.

We believe that we have adequately discussed the observed deviations of the sensitivity cases from the ambient cases. It is clear in the text that we classify all cases into the following types: ambient cases, OFR cases with strong water vapor photolysis (larger deviations), other OFR cases (smaller deviations), chamber cases without wall partitioning (very small deviations), and chamber cases with wall partitioning (substantial deviations). It is an important conclusion of this paper that differences among the cases of a precursor are generally smaller than those among different precursors. So in most situations, it was sufficient to discuss these types of cases to capture their features and discussions on a case-by-case basis were not necessary. When such discussions were necessary, especially for HO$_x$ recycling ratios, we made them for the cases that could not be characterized well by the general features of the corresponding type of cases, for example:

"*As more multifunctional species are formed (particularly through ring-opening) near the OH$_{exp}$ of the peak OHR$_{VOC}$, HO$_x$ recycling is also active, with $\beta_1$ increasing and $\beta_2$ remaining high (Fig. 1). There are a few peaks in $\beta_1$ and $\beta_2$ for certain chamber cases. The peak in $\beta_2$ for the chamber case with high initial OHR (100 s$^{-1}$) and no walls results from RO$_2$ cross-reactions, many of which produce alkoxy radicals that subsequently yield carbonyls and HO$_2$ through reactions with O$_2$ (Orlando and Tyndall, 2012).*"

Therefore, we do not modify the text to address this comment.

R2.8) In addition, it is difficult to discriminate the lines of multiple sensitivity simulations. Line styles or symbols should be adequately selected for clearer visibility.

We use more colors and fewer styles for the lines to make them more distinguishable. The updated figure is reproduced below:

[Figure]

[Figure]

**Figure 1. Total organic OH reactivity (OHR) with and without the contribution of the precursor, OH recycling ratio ($\beta_1$), and HO$_x$ recycling ratio ($\beta_2$) as a function of OH exposure (or equivalent photochemical age; second x-axis) in the photooxidations of decane, isoprene, and m-xylene under different conditions in the atmosphere, oxidation flow reactor (OFR), and chamber.**"

R2.9) Figure S4: Legend for "organic peroxy" is not shown.

Corrected. Please see the updated figure in the response to comment R2.2.

**References for the responses to both reviewers**

IUPAC Task Group on Atmospheric Chemical Kinetic Data Evaluation: http://iupac.pole-ether.fr/#.

[revised manuscript text omitted]